# Pulsatile electrical stimulation creates predictable, correctable disruptions in neural firing

Cynthia R. Steinhardt [1,2] ✉, Diana E. Mitchell[1,3], Kathleen E. Cullen [1,4] & Gene Y. Fridman [1,4]

Electrical stimulation is a key tool in neuroscience, both in brain mapping studies and in many therapeutic applications such as cochlear, vestibular, and retinal neural implants. Due to safety considerations, stimulation is restricted to short biphasic pulses. Despite decades of research and development, neural implants lead to varying restoration of function in patients. In this study, we use computational modeling to provide an explanation for how pulsatile stimulation affects axonal channels and therefore leads to variability in restoration of neural responses. The phenomenological explanation is transformed into equations that predict induced firing rate as a function of pulse rate, pulse amplitude, and spontaneous firing rate. We show that these equations predict simulated responses to pulsatile stimulation with a variety of parameters as well as several features of experimentally recorded primate vestibular afferent responses to pulsatile stimulation. We then discuss the implications of these effects for improving clinical stimulation paradigms and electrical stimulation-based experiments.

Electrical stimulation has a long history in neuroscience research as a pivotal tool for advancing our understanding of both the functional roles of localized neuronal populations and the connectivity of neural circuits. Invasive electrical stimulation has also become an increasingly popular clinical intervention to treat a wide range of neurological disorders[1,2]. Applications include restoration of sensory function[3–5] and treatment of diseases, including Parkinson's disease[6], seizures, and even psychiatric disorders[7]. Across these invasive applications, neural implant-based treatments all rely on biphasic, charge-balanced pulses to interact with the impaired neural system in order to keep current delivery safe for the target tissue at the stimulation site[8]. As a result, electrical stimulation has become synonymous with pulsatile stimulation.

While pulsatile stimulation-based treatments have successfully aided in a range of restorative and suppressive treatments[1,2], patient recovery typically remains significantly below normal levels of function; in each case, system-specific explanations have been offered, ranging from unnatural recruitment of neurons and, therefore, network-level adaptation[9], to local interference based on physiology[10,11]. Neural engineers have explored the factors that impair neural implant performance using detailed biophysical models that include neuron-specific channels, ion densities, and physiology[12]; such modeling has been especially pertinent because stimulation artifacts and technological limitations often prevent direct observation of neural responses during therapeutic intervention. Particularly, the deep brain stimulation (DBS) field has used this approach to understand the impact of parameters such as pulse waveform, electrode orientation, and tissue properties on neural activation[13]. Successes in this field have led to the use of patient-specific modeling as a popular clinical approach for finding patient-specific stimulation parameters that improve the performance of a variety of implants[11,14,15]. These parameterizations, however, do not account for another essential feature of neural responses: the neuronal firing pattern over time.

[1]Department of Biomedical Engineering, Johns Hopkins School of Medicine, Baltimore, MD, USA. [2]Center for Theoretical Neuroscience, Columbia University, New York, NY, USA. [3]Department of Neurosciences, Faculty of Medicine, University of Montreal, Montreal, QC, Canada. [4]Department of Otolaryngology, Johns Hopkins School of Medicine, Baltimore, MD, USA. ✉e-mail: cs4248@columbia.edu

Producing consistent, interpretable neuronal firing patterns in real-time is a critical factor in restoring function, particularly in sensory systems, where the natural firing patterns carry information about time-varying sensory input signals to the brain. Neural implants, therefore, employ algorithmic mappings that determine the stimulation parameters needed to evoke the desired neuronal firing pattern. Standard stimulation strategies include fixed-amplitude pulse rate modulation[16,17] and fixed-rate pulse amplitude modulation[18], where the fixed parameter is set at a high level in both cases. An assumption inherent to these fixed-parameter strategies is a consistent linear mapping between the number of stimulation pulses and neuronal firing[19]. However, experimental observations and mathematical modeling[10,20–22] have identified effects that can lead to time-varying differences in firing rate, including facilitation and blocking[10,20], especially when combined with ongoing spontaneous (natural) firing activity. We propose that these effects, which lead to complex relationships between pulse parameters and neural activation, are a common reason for the limited restorative efficacy of neural implants.

One approach for accounting for these complicating effects is to include detailed biophysical simulations within the neural implant algorithms and models; however, this approach is computationally intensive and presently intractable. Here, we take a different approach to this question: we use a detailed biophysical model to investigate factors of spontaneous activity and pulse parameterization that impact firing rate and extract general principles of pulsatile interactions from the simulation. We use these rules to generate time-independent equations that can estimate the induced firing rate in response to pulse parameters and could be parameterized for various neuronal systems based on measurable observations of the system. An advantage of this approach is that resulting equations can be inverted and integrated into real-time devices to correct for complex effects of pulses on firing rate in a computationally efficient way, improving our ability to precisely control neural firing rate over time.

In this paper, we work towards these goals by studying the range of effects of pulsatile stimulation on vestibular afferents. We choose vestibular afferents because they have a large range of spontaneous activity and neural firing regularity that can be used to probe the causes of variability in response to pulsatile stimulation[23]. We use a validated vestibular afferent model[24–26] that can be tuned to capture vestibular afferent-specific properties to explore the effects of pulse rate, pulse amplitude, and spontaneous activity on induced neural firing. We find a diversity of effects that can be understood as two phenomenological categories of interactions: pulse-pulse interactions and pulse-spontaneous interactions. We create time-independent equations that capture these effects and show that they fit the simulations and align with re-analyzed experimental data. Finally, we assess the applicability of these equations to pulsatile modulation paradigms.

## Results

### Identifying complex effects of pulsatile stimulation
Previous experimental recordings of vestibular afferents indicate that electrical stimulation pulses produce variable numbers of action potentials under standard experimental and clinical conditions (Fig. 1)[9,27]. To understand the complexities of pulsatile stimulation, we use a detailed biophysical model of the vestibular afferent to simulate afferents with different spontaneous firing rates and the effect of pulses of varying pulse amplitude and pulse rate on their firing rate ($F$). Throughout the text, we use the term spontaneous to distinguish naturally occurring activity, meaning excitatory post-synaptic currents (EPSCs) and ESPC-induced spiking, from pulse-induced spiking. Based on these observations, we show that pulses have two categories of interactions: pulse-pulse interactions, effects on channels that change the probability of other pulses producing action potentials (APs), and pulse-spontaneous interactions, effects on channels that change the probability of spontaneous EPSCs making APs and vice versa. We

develop time-independent equations that capture the facilitation, additive, and blocking effects within each category of interaction. The equations are fit to simulate afferent responses to fixed-rate, fixed-amplitude pulsatile stimulation across all stimulation parameters. We show these equations comply with observations from re-analyzed experimental vestibular afferent recordings and that they can be extended to predict responses to modulated waveforms of pulsatile stimulation on the millisecond timescale.

In Mitchell et al.[9], extracellular recordings of individual vestibular afferents were made in response to one-second blocks of fixed-pulse rate ($R$), fixed-pulse amplitude ($I$) extracellular stimulation. Pulse rates were varied between 25 and 300 pps, while pulse amplitude was fixed. Based on the intuition that a suprathreshold pulse (80% of the level of facial twitch) will induce an AP, at suprathreshold $I$s, the pulse rate-firing rate relationship (PFR) is expected to be $F = R$ at all $R$s. Instead, the best linear fit of the PFR has a slope less than 1 across afferents, with the highest PFR slope being $F = R/2$ in an afferent with a spontaneous rate($S$) of $43$ sps (Fig. 1a–c black).

We simulate individual vestibular afferents using a modified version of the biophysical model developed by Hight and Kalluri which we used in previous studies[25,26]. The channel conductances are tuned to match firing regularity, and the inter-EPSC interval ($\mu$) is tuned to match the recorded spontaneous firing rate (Fig. 1b). At 230 μA, the simulated afferent produces a PFR that closely matches experimental observations from the afferent with the highest PFR slope ($N = 50$, rms = 11.4 ± 4.6 sps, Fig. 1b red). We use this model to explore the variety of PFRs produced with different pulse parameters, under the assumption it will exhibit the largest range of PFRs.

We conduct a full sweep of $I$s from 0 to 350 μA and $R$s from 0 to 360 pps in steps of 12 μA and 1 pps and simulate responses to one-second blocks of pulses with each combination of parameters (Fig. 1d). Instead of the PFR linearly increasing with $R$ (black), multiple bends occur in the PFR. Only for a small subset of parameters is $F = R$. The maximum increase in firing rate is lower with higher $S$s and the highest $R$s. Additionally, $I$ did not only have a strong additive effect above a threshold level. At higher $I$s, even spontaneous activity was blocked in afferents of all $S$s (Fig. 1e). Based on our previous work[26], we hypothesize that the non-linearities derive from two simultaneous interactions that occur during pulsatile stimulation at any axon: pulse-pulse and pulse-spontaneous interactions. To isolate the contribution of each type of interaction, we perform simulations with no EPSC activity and characterize the effects of pulses alone on the axon (pulse-pulse interactions). Then, we reintroduce EPSCs into the model and characterize pulse-spontaneous interactions.

### Pulse-pulse interactions
Once all EPSCs are removed from the model, we introduce the same set of fixed-rate fixed-amplitude pulsatile stimulation blocks and observe a smooth transition between three stages of effects as pulse amplitude increases: facilitation, addition, and suppression (Fig. 2a–c). These effects result from the fact that pulses produce unnatural perturbations to the channel states of the voltage-gated ion channels by creating changes in membrane potential of atypical amplitude and duration.

The pulse-induced changes in membrane potential produce effects analogous to EPSCs but over different length time windows that depend on pulse parameters. Pulse-pulse facilitation (PPF) is analogous to natural facilitation. At a subthreshold pulse amplitude, $R$ must exceed some rate for pulses to additively increase the membrane potential and produce an AP (Fig. 2d1). As the $I$ increases, the pulse rate at which a pulse is sufficient to create APs ($R_{ppfacil}$) shifts towards 1 pps, and the number of pulses required to produce APs shifts to one (Fig. 2a). This can be modeled with a sigmoid function of height and center dependent on $I$ ("Methods" section; Eq. 14).

Once $I$ exceeds a threshold amplitude, all pulses cause pulse-pulse addition (PPA) and pulse-pulse blocking (PPB; Fig. 2b). In this pulse

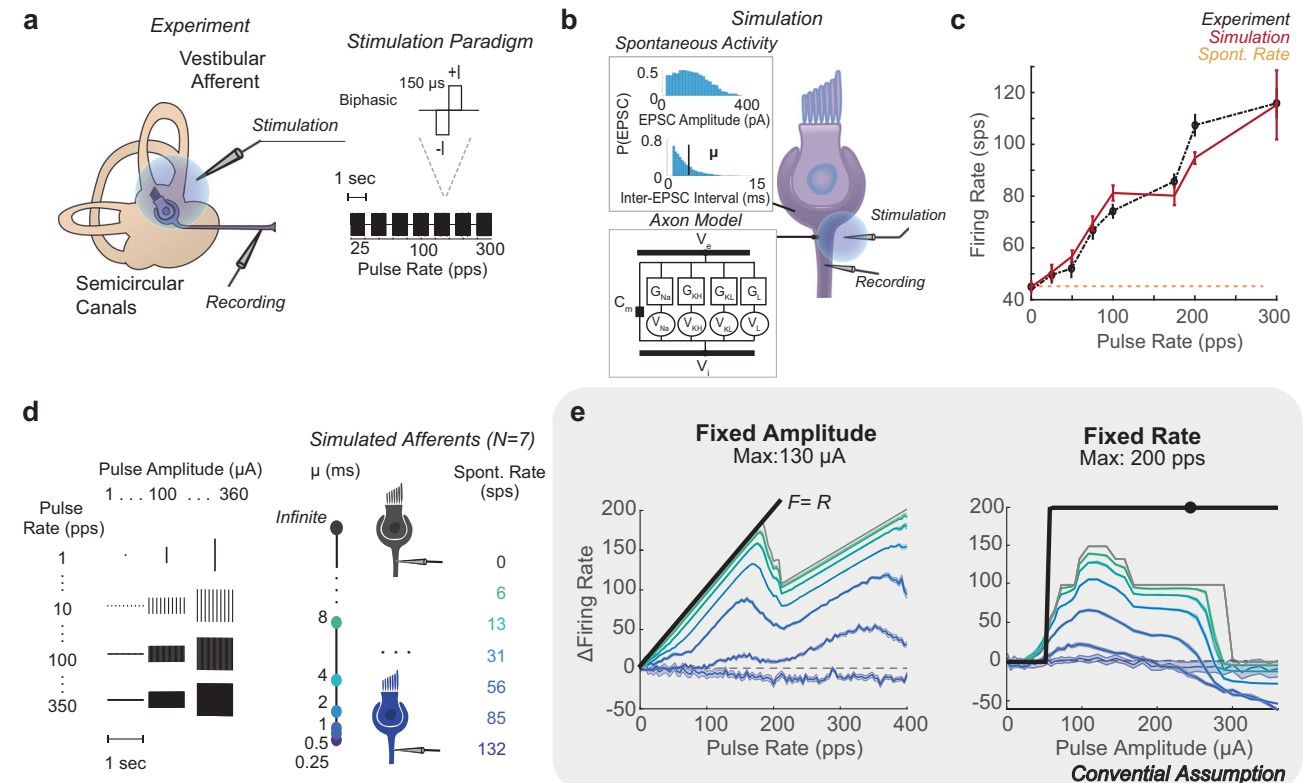

**Fig. 1 | Demonstration of Pulsatile Effects. a** In the Mitchell et al. experiment, individual vestibular afferents were recorded in response to biphasic pulse blocks of 1 s of pulse rates from 25 pps to 300 pps. **b** Individual afferents were simulated with a Hodgkin-Huxley style model of axonal channels and membrane potential changing with EPSCs of a given amplitude and rate, where inter-EPSC interval μ is tuned to match spontaneous firing rate. The model was tuned to match the irregularity and spontaneous rate of the experimental afferent. **c** The pulse rate-firing rate relationship (black) at a fixed pulse amplitude of a recorded afferent with a spontaneous firing rate of 43 sps (yellow). At a simulated pulse amplitude of 230 μA, the pulse rate-firing rate relationship closely matched experimental data

(red, $N = 50$ stochastic simulations with $I = 230\,\mu A$). **d** A full sweep of pulse parameters from 0 to 350 pps and 0 to 360 μA in steps of 1 for 1-second blocks was simulated on models of afferents with different spontaneous rates, as controlled by varying μ. **e** This revealed a complex relationship between pulse parameters, spontaneous rate, and firing rate. Maximum change in firing rate is shown for the model of afferents with different spontaneous firing rates in the same colors as **d**. Conventional assumptions of pulse parameter relationships to change in firing rate are shown in black. Maximum response is significantly lower than expected as pulse rate (left) and pulse amplitude (right) vary. Data are plotted as mean ± SEM across simulations in **c** and **e**.

amplitude range, pulses produce changes in membrane potential large enough to create APs but also create significant artificial after-hyperpolarizations that can block following pulses from producing APs. We call the length of time after a pulse in which a following pulse would be prevented from making an AP $t_b$, or the block time. Due to the refractory effects, at low $R$, the $F$ falls on the line $F = R$ (Fig. 2d2), but, as $R$ increases past $R_b = 1/t_b$, the inter-pulse interval becomes less than $t_b$. At $R > R_b$, after a pulse makes an AP, the next pulse delivered arrives within $t_b$, preventing an AP or the full refractory period that would follow an AP from being formed (Fig. 2d3.1). As a result, the third pulse in the sequence produces an AP again. This pattern repeats throughout the stimulation time, leading to a relationship of $F = R/2$. Throughout the additive pulse amplitude range, the PFR starts as $F = R$ and drops from the line $F = R/n$ to $F = R/(n+1)$ as $R$ increases above $n/t_b$, where $n = 1,2,3…$ is the number of pulses blocked before another AP is made (Fig. 2b). We model the resulting firing rate as $\frac{R}{\lceil t_b R \rceil} = \frac{R}{\lceil R/R_b \rceil}$.

The PFR does not transition directly from $F = R/n$ to $F = R/(n+1)$ at $R_b^n = n/t_b$. Instead, the PFR has a bend, where the slope of the PFR decreases smoothly from 1 sps/pps, starting at $R_{pb}$ (open circle), a $R$ less than $R_b$, to 0.5 sps/pps at $R_b$ (closed circle, Fig. 2e). In this range of $R$, pulse-pulse partial block occurs. This effect resembles facilitation. Inter-pulse intervals are short enough for refractory effects to build, but these interactions build to one pulse in a sequence of three or more pulses being blocked instead of one in the sequence producing an AP (Fig. 2f, lime green). Pulse-pulse effects arise from voltage changes affecting the opening and closing of a combination of axonal

voltage-gated channels, but a correlate of the effect on the axon state can be observed in the sodium channel dynamics. Here, the $m$-gate reducing with each pulse (gray), shows the building-blocking effect (Fig. 2f circles). Although a sequence of pulses producing partial block may produce a complex pattern of blocked and added APs, we can estimate the effect on average as the probability of the next pulse in the sequence arriving and being blocked gradually decreasing from 0 to 1 between $R_{pb}$ and $R_b$ (Fig. 2e).

We incorporate the partial block effect in the $\psi(I, S, R)$ term, which captures how the length and falloff of this partial-block window changes with pulse parameters. The ongoing spontaneous activity creates additional resistance to the pulse, changing the membrane potential so $\psi$ has an additional dependence on $S$ (as discussed in the next section). This leads to our estimate of the firing rate produced by pulse-pulse effects (excluding the facilitation window), $F_{pp}$:

$$F_{pp} = \frac{R}{\lceil R/R_b \rceil + \psi(I, S, R)} \tag{1}$$

$$\psi(I, S, R) = \sum_{n=1}^{N} \psi_n(t_b, k_{pb}^n, t_{pb}^n) \tag{2}$$

where $\psi(I, S, R)$ is a sum of partial-elimination effects per bend $\psi_n$ in the PFR where $n$ is the bend number that shows PPB at $nR_b$. $\psi_n$ transitions from 0 to 1 creating the smooth falloff between $R_{pb}$ and $R_b$. Scaling of

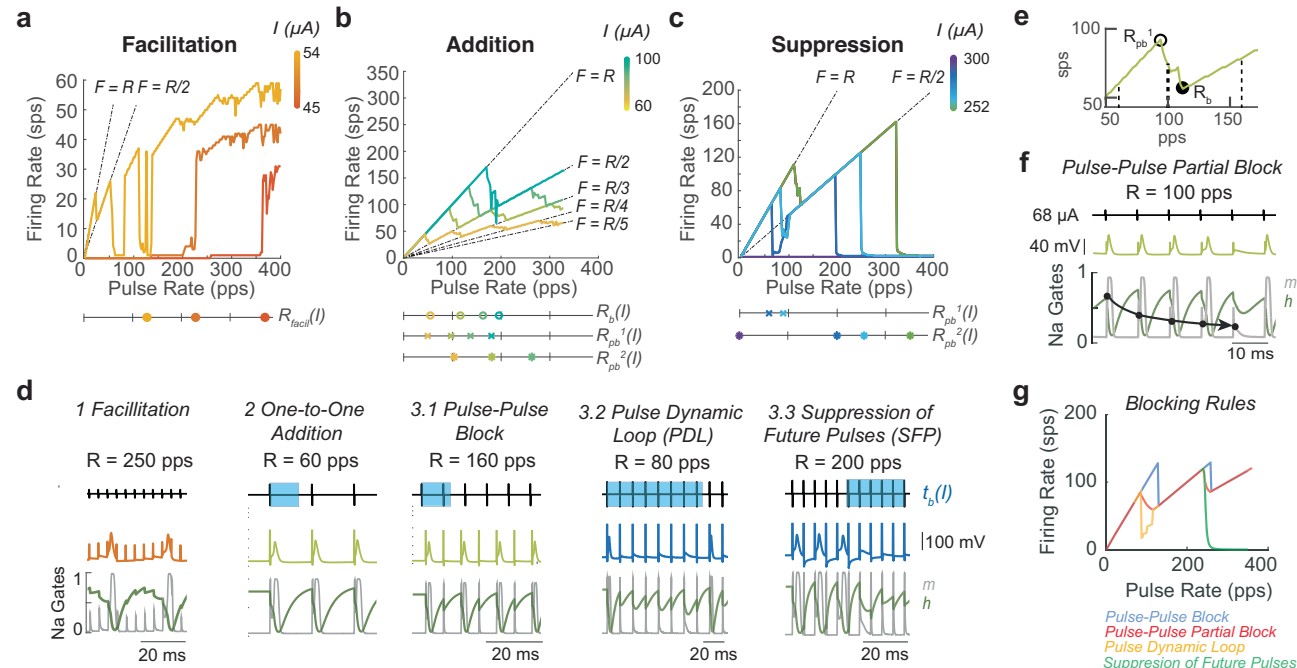

**Fig. 2 | Pulse-pulse interactions.** Interactions go through three stages: (**a**) facilitation, (**b**) addition, and (**c**) suppression. In each stage, as current increases (indicated by color bar in panels **a**–**c**) the pulse-pulse interactions change, changing in the PFR. The parameters governing the dominant effect are shown on the line graphs below each figure. Colors of markers on line graphs are matched to $I$ in color bars. The lines $F = R/n$, $n = 1,2,3…$ are shown with dashed lines on graphs to compare to local slope of the PFR. **d** Examples of each effect are shown with the pulse train on the top. The duration of the full-block window $t_b$ is marked in blue on the pulse train. The voltage trace is shown with color matched to the current level that produces them in the PFR plots in **a**–**c**. The dynamics of the h-gate (green) and m-gate (gray) of the sodium channel are shown. The dynamics can be used as a correlate of the axon state. **e** A close-up of the PFR for 68 µA (the $I$ where pulse-pulse block was demonstrated in d2 and d3.1, lime green from **b**). $R_b$ (filled circle) and $R_{pb}^1$ (open circle) are marked on the PFR. The Rs at which effects from d are shown(thin dash line), as well as example $R = 100$ pps where partial block occurs (thick dash line). **f** Partial block is illustrated overtime at 68 µA in the same format as d. In m-gate(gray), reduced channel activation can be seen (marked with circles) with each pulse, leading to 1/5 pulses per sequence being blocked ($F = 0.8R$ in **e**). **g** Comparison of the rules for each pulse-pulse blocking effect for the same parameters $t_b$, $p_{bp}^{1/2}$, and $k_{bp}^{1/2}$ are shown and colored by the implemented rule.

blocking at the bends $k_{pb}^n$ and $p_{pb}^n$, where $R_{pb}^n = p_{pb}^n R_b$, the fraction of $R_b$ from which partial-block beginning on the PFR are the driving parameters that depend on $I$, $S$, and $R$. This effect is bend-specific because the partial elimination zone becomes shallower and narrower at higher firing rates. For details, see Methods Eqs. 10–13.

Under most conditions, $\psi(I, S, R)$ represents a smooth transition in the likelihood of pulses creating APs from $1/n$ to $1/(n + 1)$ at every bend (Fig. 2b) that we refer to as the standard pulse-pulse blocking effect. However, as the pulse amplitude increases from the one-to-one additive zone into the suppression zone, blocking effects extend beyond blocking individual pulses, leading to two exceptional versions of the $\psi_n$ term: pulse dynamic loop (PDL, $\psi'_1$) and suppression of future pulses (SFP, $\psi'_2$) (Fig. 2c, "Methods" section; Eqs. 12–13). SFP occurs when $R > 2/t_b = R_b^2$. Instead of every third pulse creating an AP, the afterhyperpolarization effects compound and hold channel dynamics in a state where they cannot reopen, which causes the firing rate to quickly drop to zero (Fig. 2c, d). As $I$ increases, the second bend ($2R_b$) moves towards 0 pps, until all activity is suppressed and $F_{pp} = 0$ (Fig. 2c right, d3.3). PDL is a version of this effect that occurs around $R = t_b$ in which pulse timing is aligned to harmonics of the channel dynamics so that the channels cannot recover until many pulses occur, but, at higher $R$s, this loop is broken and $F = R/2$ holds (Fig. 2c right, d3.2).

Transitions through each of these effects at different $I$s can be seen in Fig. 2a–c as well as how parameters $R_{ppfacil}$, $R_b$, and $R_{pb}^{1/2}$, change with $I$. Each of these effects can be visualized over time in Fig. 2d for a single $I$ from Fig. 2a–c and a single $R$. They are shown compared to $t_b$ and the dynamic gates of the sodium channel, where the time where $h$-gate is at an intermediate value is highly correlated to the axon dynamics becoming blocked. It is important to note that these effects result from the dynamics of a system of non-linear dynamics equations, but, due to the regularity of pulse timing and the perturbation to the channels by the fixed pulse amplitude, these effects can be estimated over time with time-independent parameters $t_b$ and $p_{pb}^n$, resulting in equations that can capture each of these effects. Figure 2g shows the prediction equation for each of the blocking effects with equivalent parameters. With each of these effects characterized and parameterized, we turn to the pulse-spontaneous interaction. For a summary of how each parameter changes with pulse amplitude and affects the PFR see the left side of Supplementary Fig. 1.

### Pulse-spontaneous interactions

To characterize how PFRs change with pulse parameters and spontaneous rate, we test the response of simulated irregular vestibular afferents with spontaneous rates from 6 to 132 sps (the full span of natural spontaneous rates observed) to the same pulse parameters used during pulse-pulse interaction testing(0–360 µA, 0–350 pps). We create afferents with six different spontaneous rates within the natural range and test their response to pulses across 10 trials with each combination of pulse parameters. This allows us to account for variability across simulations due to the stochastic EPSC timing.

To capture how pulsatile stimulation produces non-monotonic PFRs in the presence of spontaneous activity, we create equations that estimate the contribution of pulse-pulse interactions to firing rate ($F_{pp}$) and pulse-spontaneous interactions to firing rate ($F_{ps}$). These terms can also be re-arranged to estimate the contribution of spontaneous APs and pulse-induced APs to $F$ separately (see "Methods" section). Prior work by our group and others has attempted to capture these interactions using simplifying equations[26,28], but those attempts do not

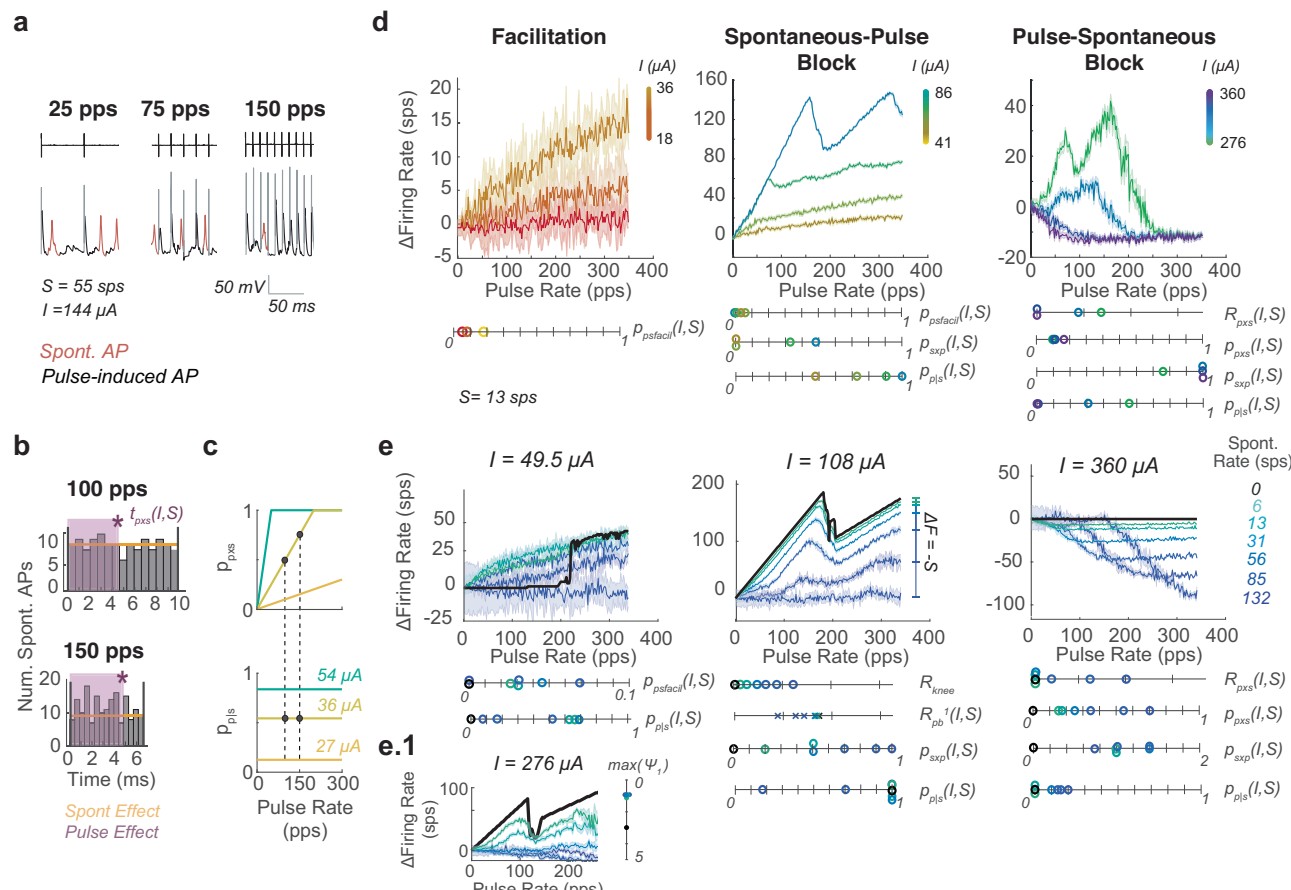

**Fig. 3 | Pulse-spontaneous interactions. a** Simulated traces during stimulation with spontaneous activity where spontaneous APs are shown in red and pulse-induced APs in black. Artifacts are grayed for clarity. For the same pulse amplitude, at different pulse rates, the presence of each type of AP differs. **b** In the time between pulses spontaneous activity is approximately normally distributed (gray). Thus, it is estimated as uniformly distributed(yellow). As pulse rate increases for the same pulse amplitude stimulating the same afferent, the same length of blocking effect $t_{pxs}(I,S)$ (purple)is present, but with a shorter interval between pulses. **c** Thus, the probability of a spontaneous AP being blocked by a prior pulse ($p_{pxs}$) increases linearly with $R$ until it reaches 1, all pulses blocked (top). Meanwhile, for a given amplitude the probability of a pulse being blocked due to spontaneous activity ($p_{p|s}$) is the same as pulse rate increases due to equally distributed spontaneous activity (**b**) (bottom). Two dots at $I = 36$ μA indicate $R$ and $I$ combination leading to histograms in b. The level of blocking plotted with changes in $I$ (colored lines). **d** Pulse-spontaneous interactions evolve through a facilitation, spontaneous-pulse block, and pulse-spontaneous block zone as pulse amplitude increases which co-exists with the pulse-pulse blocking rules. Parameters ($p_{psfacil}$, $p_{pxs}$, $p_{sxp}$, $p_{p|s}$) governing each are shown changing with pulse amplitude for cases where $S = 13$ sps. Pulse amplitude colors are on the same scale as in Fig. 2e) The same categories of rules and their parameters are shown changing with spontaneous firing rate (colors on right). When a parameter is unlisted below, assume all values are zero across shown cases. **e.1** The spontaneous-pulse blocking effect suppressing pulse dynamic loops. Data are plotted as mean ± SEM across simulations in **d** and **e**.

provide a complete description of the effects observed in our simulation described below.

For a fixed $I$, as $R$ increases, the presence of pulse-induced and spontaneous APs changes (Fig. 3a). Although EPSC timing and thus the subset of EPSC events that generate APs are stochastic, because of their frequency compared to pulses, interactions can be estimated to occur with approximately uniformly distributed EPSCs (Fig. 3b yellow line, histogram). For a given $I$, there is some $t_{pxs}(I,S)$ after a pulse for which a pulse blocks EPSCs from becoming APs(purple), an analogue of $t_b$. As $R$ increases, the ratio of $t_{pxs}(I,S)$ to the inter-pulse interval ($1/R$) linearly increases to 1; we capture this effect with $p_{pxs}(I,S)$, the probability that a pulse blocks spontaneous APs, where once $p_{pxs}(I,S) = 1$, each pulse blocks all spontaneous APs in between (Fig. 3c top).

At the same time, the ever-present EPSCs create a constant resistance of the axon to pulses, captured by $p_{p|s}$, the probability that a pulse produces an AP given the spontaneous activity level. When $I$ is low, $p_{pxs} = 0$ and $p_{p|s} = 0$. As $I$ increases, pulses are sufficient to overcome the EPSC activity and eventually block all spontaneous APs, so $p_{pxs}$ and $p_{p|s}$ go to 1 (Fig. 3c, d and Supplementary Fig. 2 for changes with $I$ and $S$). This picture of increasing interaction as $R$ increases (Fig. 3b) can be used to visualize why $p_{psfacil}$, the probability of facilitation between pulses and EPSCs, also increases linearly with $R$ at low $I$s. A similar picture applies for $p_{sxp}$ the probability that EPSCs block pulses from becoming APs. Pulses segment time into inter-pulse intervals, and there is a probability within those intervals of EPSC activity capable of blocking pulses occurring just preceding the pulse, leading to the pulses being blocked. These blocking effects that linearly increase with $R$ co-occur for a majority of $I$s, making them difficult to isolate in the PFR plots. As such, we show the relevant combination of parameters and their scale below plots in Fig. 3d, e and the line graphs below to elucidate how $I$ and $S$ affect those parameters separately. Additionally, in Supplementary Fig. 1, right, we highlight the effects of $p_{psfacil}$, $p_{pxs}$, $p_{sxp}$, and $p_{p|s}$ on features of the PFR as $S$ increases. Each isolated effect is plotted in red over a PFR trace in insets to the right of the main plots for clarity.

All these effects sum to produce $F_{ps}$, the contribution of pulse-spontaneous interactions to firing rate:

$$F_{ps} = p_{psfacil}R + \max\{-Sp_{p|s}, -p_{p|s}p_{sxp}R\} + \max\{-S, -p_{pxs}(R - R_{pxs})\} \quad (3)$$

Spontaneous-pulse(SP) blocking is only observed to block up to one pulse per spontaneous AP in this pulse parameter range, leading to the $\max\{-Sp_{p|s}\}$ term in Eq. 3, where at $p_{p|s} = 1$, $S$ pulses are blocked. The SP blocking term is scaled by $p_{p|s}$ because the presence of blockable pulses is scaled down but evenly distributed throughout time, leading to a scaled reduction in pulses for all $R$. We also observe that, during pulse-spontaneous (PS) blocking, pulses self-facilitate and initially start blocking spontaneous APs only at high $R$. So, we add the term $R_{pxs}$ that shifts the $R$ at which PS blocking starts (Eq. 3 and Supplementary Fig. 1).

Similar to the pulse-pulse interactions with which these effects co-occur, pulse-spontaneous interactions smoothly transition through three stages as $I$ increases. At low $I$, only facilitation occurs, so $p_{psfacil}$ increases. Once $I$ is large enough, $p_{p|s}$ approaches 1, and $p_{psfacil}$ goes to 0, while spontaneous-pulse blocking begins, reflected by $p_{sxp}$ growing. Then, at high $I$, pulse-spontaneous blocking dominates, captured by $p_{sxp}$ growing (Fig. 3d). As $I$ increases, $R_{pxs}$ shifts left and $p_{sxp}$ increases until $F = 0$ at all $R$ (Fig. 3d, e and Supplementary Figs. 1 and 2).

These effects also depend on $S$, as shown in Fig. 3e–e1. The dominant effect is that as $S$ increases the axon becomes more resistant to pulses. So, at $S = 132$ sps, we observe almost no facilitation, nearly no pulse-pulse addition, and a blocking effect that starts at a larger $R_{pxs}$ for the same $I$ and requires larger $I$ to drive $F$ to zero (Fig. 3e dark purple on line graphs). Facilitation is a slight exception in that $p_{psfacil}$ increases with $S$ until a threshold level of spontaneous activity ($S > 60$ sps) above which primarily SP blocking occurs (Fig. 3d left green vs. blue traces and circles, Supplementary Fig. 1). At midrange $I$ (center), $p_{sxp}R$ reaches $S$, the maximal effect, for all $S$ cases (Fig. 3e middle). This leads to a bend in the PFR ($R_{knee}$) which is the lowest $R$ that satisfies $p_{sxp}R_{knee} = S$. At $R > R_{knee}$, the PFR is shifted down by $S$, leading to almost no $\Delta F$ at higher $S$ because $F_{pp} < S$ at most $R$s (Fig. 3d middle). The point where $R_{knee}$ would have been visible may not be present in PFRs at high $S$ (as in at $S = 132$ sps, $I = 108$ μA). This is due to the combination of the low increase in firing rate with pulses ($F_{pp}$) and the strong blocking effects blocking all addition of pulses. Mathematically, this is captured in the $\max\{-Sp_{p|s}\}$ term that described the observed limitation to blocked APs. $R_{knee}$ could still be predicted as it is in Fig. 3e. At high $I$, the combination of high $I$ pulses and EPSCs together add to create SFP that blocks pulses, so $p_{p|s}$ returns to 0, and $p_{sxp}$ goes to 1. In this $I$ range (right), as $S$ increases, PS blocking starts at a higher $R_{pxs}$ and reduces $F$ less, because it requires higher $I$ pulses to cause equivalent levels of blocking as with lower $S$ afferents (Fig. 3e right, Supplementary Fig. 1).

Finally, $S$ also affects the partial block window of pulse-pulse effects. EPSCs act as a level of noise correlated to $S$, which extends recovery of the axon after pulses, increasing $p_{pb}^n$ (Fig. 3e middle). Spontaneous activity also prevents PDL by causing too much noise for channels to remain in a dynamic loop so that $\psi_1$ never exceeds 1 (Fig. 3e.1). Example traces of the pulse-pulse effects occurring in afferents with different $S$ are shown, like in Fig. 2d, in Supplementary Fig. 3.

The induced firing rate ($F$) can be estimated as the combination of $F_{pp}$, $F_{ps}$, and $S$:

$$F = \max\{0, p_{p|s}F_{pp} + F_{ps} + S\} \quad (4)$$

where $p_{p|s}$ scales down $F_{pp}$ during SP blocking. For a visualization of how each variable changes with pulse parameters and spontaneous rate, see Supplementary Fig. 1.

## Applications of pulsatile interaction rules

We test the accuracy of these equations by parameterizing them with values that best minimize the rms error between the PFR of the simulation at fourteen current amplitudes across the seven spontaneous firing rate cases. The parameters are then interpolated for the thirty held-out current amplitude conditions across afferents. We find that the equations (red) closely approximate the complexity of the PFRs across conditions (Fig. 4a blues). The rms error averages $5.77 \pm 1.19$ sps across all fits ($N = 44$, Supplementary Table 2), and there is no significant difference in the fit of the parameterized and interpolated conditions, indicating smooth, precise parameterizations could be found (Fig. 4b and Supplementary Fig. 2). We note relative variability in fits at low $S$ compared to high. One source of variability is an accumulation of error at the sharp drops during PP blocking (Fig. 2c), due to the parameters of our equations being bounded to keep parameter exploration reasonable. While, at high $S$, pulses contribute few APs so non-monotonic blocking effects (PPB, SFP, etc.) are low amplitude, and linear PS and SP blocking effects dominate, which are easily fit with linear rules. Still, this rms level is also less than the standard deviation across 10 simulation runs with different seeds for some afferent conditions (Fig. 4c). We assess sensitivity of fit to each parameter, revealing that, although each parameter influences the PFR (Supplementary Fig. 1), particularly $t_b$, $p_{pb}^{1/2}$, and $p_{p|s}$ have strong influence on error in the PFR (Supplementary Fig. 4); the pulse-pulse parameters affect rms more with no spontaneous activity. However, as $S$ increases, the various pulse-spontaneous parameters have similar levels of influence to other parameters (Supplementary Fig. 4).

We then test whether these equations reflect observable features of experimentally recorded vestibular afferents. We reanalyze recordings from six afferents from the Mitchell et al.[9] study, which focused on central adaptation but recorded vestibular afferent responses to pulsatile stimulation at multiple amplitudes. This provided 5 afferent recordings at the maximum safe pulse amplitude and 4 afferent recordings at 18 pulse amplitudes from 25% to 100% of the safe pulse amplitude range for that electrode position (see "Methods" section, all data in Supplementary Fig. 5). The PFRs show non-monotonicities that could be explained by PPB effects, SFP at high $R$s, and changes in PFR with $I$ that reflect results of the simulations (Fig. 4d and Supplementary Fig. 5a, b).

The experimental PFRs could fit with the equations described above. However, the sparsity of pulse rate and pulse amplitude sampling causes multiple parameterizations of our equations to result in a low rms fit, making it unclear which rules shown led to the result. Instead, we use two metrics to assess the presence of the pulse effects in the data that allow data to be pooled across afferents, increasing the sample size for statistical comparisons. The slope between sampled combinations of pulse rate and firing rate (gray dash and circle) (Fig. 4d left) is used as the main metric for assessing the presence of blocking effects. The normalized area under the curve (AUC) for the PFR (Fig. 4d gray filled) is used as a metric of the level of activation (see "Methods" section). Due to PPB, we expect a higher frequency of slopes of 1,1/2,1/3, particularly at low $R$, and higher frequencies of slopes close to or less than zero due to pulse-spontaneous block, spontaneous-pulse block, and SFP. We first compare the presence of all slopes in the data to slopes in the model. To make a fair comparison to the model, we sparsely sample the simulated PFRs and slopes (see "Methods" section), producing PFRs and pulse rate-slope plots that closely resemble those sampled from experimental afferents of matched spontaneous rates (Fig. 4b right and Supplementary Figs. 5 and 6). The probability density functions of the simulated and experimental data show similar clustering around slopes of 0 with peaks forming near 0.5 and 1 sps/pps that occurred at similar pulse rates (Fig. 4e and Supplementary Figs. 5 and 6). The simulated and experimental distributions are not statistically significant (Welch $t$-test: ($t(622) = 0.31$, $p = 0.75$; Kolmogorov-Smirnov test: $p_{KS} = 0.16$). A Wasserstein distance $W(P_{exp}, P_{sim}) = 0.239$ indicates curves are close to each other. The Kolmogorov-Smirnov and Wasserstein distance statistics are significantly different than those between the experimental data and slopes derived from 5000 permutations of the

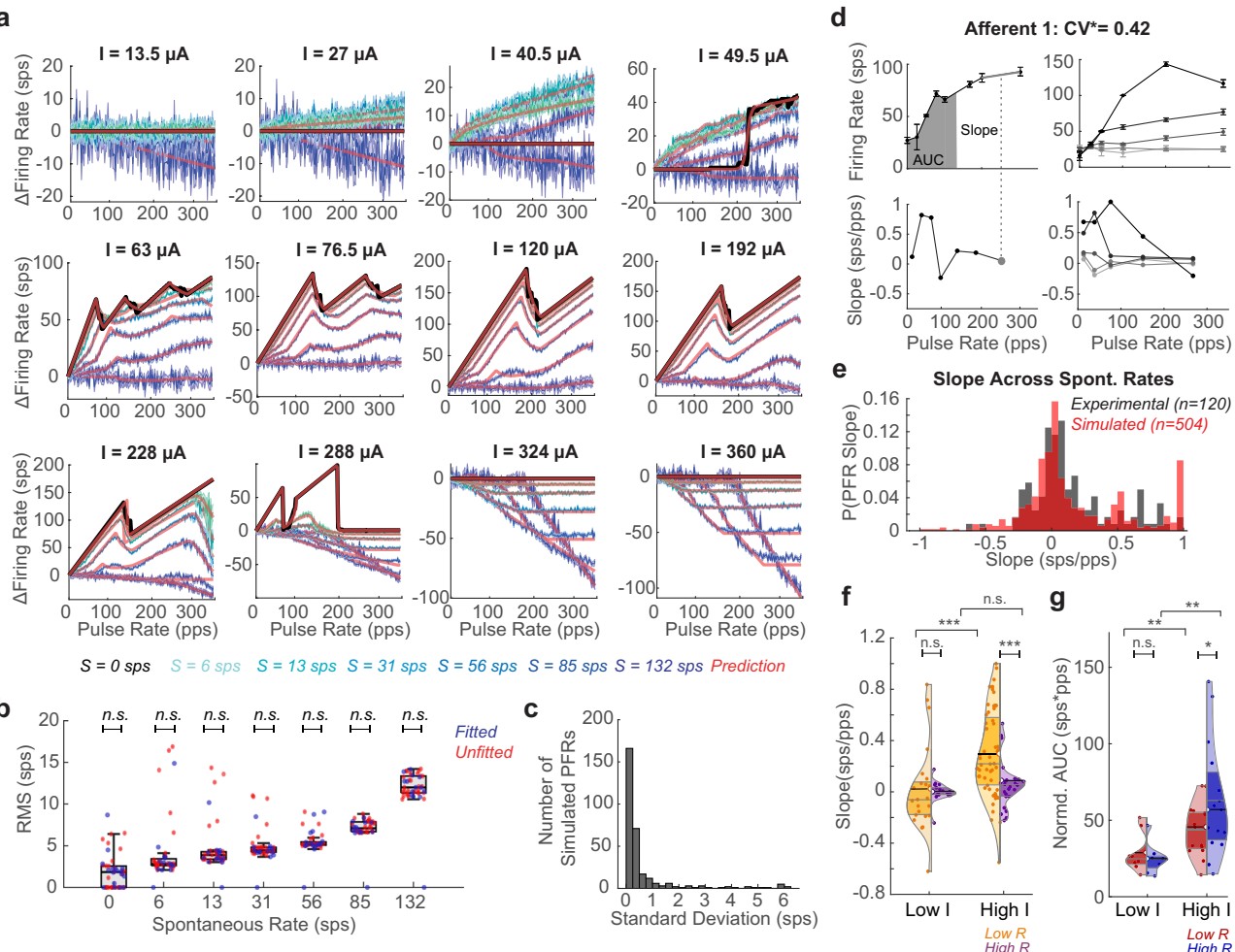

**Fig. 4 | Testing prediction equations against data. a** Fit of the equations (red) to PFR of vestibular afferents of different spontaneous rates (colored like to dark blue as *S* increases. *S* = 0 in black. Data plotted as mean ± SEM across simulations. **b** RMS between prediction and simulation for afferent models (*N* = 10) of each spontaneous rate. PFRs where parameters were fit directly (blue) and interpolated between the fitted parameters (red). Unpaired *t*-test showing no significant difference between the two (n.s.). Exact values in Supplementary Table 2. Minimum, maximum, and median (center) across amplitudes shown. **c** RMS per PFR across all simulations with different random seeds. **d** Experimental recordings from Afferent 1 (left). PFR at single I with slope highlighted on PFR and corresponding slope on pulse rate-slope plot (gray dot). AUC of firing rate at low *R* (*R* < 150 pps) also shown (gray fill). (right) PFR and pulse rate-slope plot at increasing Is for the same afferent. Data are plotted as mean ± SEM across simulations. **e** PDF comparison of experimental slopes (collected as in **d**, black) to slopes of simulated afferents sampled every 30 pps at seven current steps between 18 and 312 μA (red).

**f** Violin plots of distribution of slopes at low *R* (orange) versus high *R* (*R* > 150 pps, purple) for stimulation with high Is ($I > 0.5I_{max}$) versus low Is per afferent. At low *I*, slopes are not significantly different between low *R*(*N* = 23) and high *R*(*N* = 17) ($t(38) = 0.237$, $p = 0.81$). At high *I*, slopes are significantly different between low *R*(*N* = 54) and high *R*(*N* = 26)($t(78) = 3.32$, $p = 0.0014$). Mean marked with black line and color-matched semicircle. Median and quartiles marked with gray lines. At low versus high I, slopes at low *R* are significantly different but not at high *R* ($t(75) = 3.23$, $p = 0.0028$; $t(41) = 1.19$, $p = 0.24$). **g** Violin plots of normalized AUC in low *R* and high *R* section of the PFR for stimulation at low *I*(red) and high *I*(blue). At low *I*(*N* = 8), AUCs in the low and high *R* region of the PFR are not significantly different, but at high *I*(*N* = 15) the AUCs are significantly different ($t(14) = 0.654$, $p = 0.523$; $t(28) = 1.79$, $p = 0.0083$). All the significances reported are for two-tailed t-tests. Significance in f-g is shown as *$p < 0.1$;**$p < 0.05$;***$p < 0.01$ (Supplementary Table 3).

pulse rate-firing rate pairings across recordings, further supporting the similarities in the structure of the experimental and simulated PFRs ($p = 0$, $p = 0.007$, Supplementary Fig. 4d).

We also investigate pulse rate and pulse amplitude effects in the data. Data cannot be pooled by the *I* delivered at the electrode because the distance between an afferent and the electrode (which is not known in our experiments) affects the current level received by the afferent. We observed that experimental *I* values were only increased in a range that led to increasing activation(Supplementary Fig. 5b, c), so we assume the maximum *I* ($I_{max}$) used would be equivalent in our simulation mapping to 250 μA > *I* > 70 μA. With this assumption, we split PFRs into low *R* (*R* < 150 pps) and high *R* sections and compare their slopes at low *I* ($I ≤ 0.5I_{max}$) and high *I* to look for pulse amplitude-related effects (see Fig. 4f, g for all statistics). At low *I*, low *R* slopes are

<0.8, primarily clustering close to zero in the violin plot, which would be expected from both types of facilitation and SP-blocking and not significantly different than at high *R* (Fig. 4f left). At high *I*, we expect low *R* slopes to mostly range from 0–1 (excluding the downswing of the bend that may be captured) and high *R* slopes to cluster at negative or 0 sps/pps. We see this significant difference in the distribution of slopes ($t(78) = 3.32$, $p = 0.0014$): positive-valued low *R* slopes (orange) with clustering around 1, 0.5, 0.25–0.33. and 0 that reflects slopes from PPB and primarily zero and negative valued high *R* slopes with some samples around 0.5 and 0.3 (purple; Fig. 4f right). The differences in slope at higher *I* for the low *R* region of the PFR are highly significant ($t(75) = 3.23$, $p = 0.0028$). At low *I*, the normalized AUC of the PFR is not significantly different at low *R* or high *R*, but, at high *R*, both halves of the PFR show significantly more activation, and the high *R* portion of

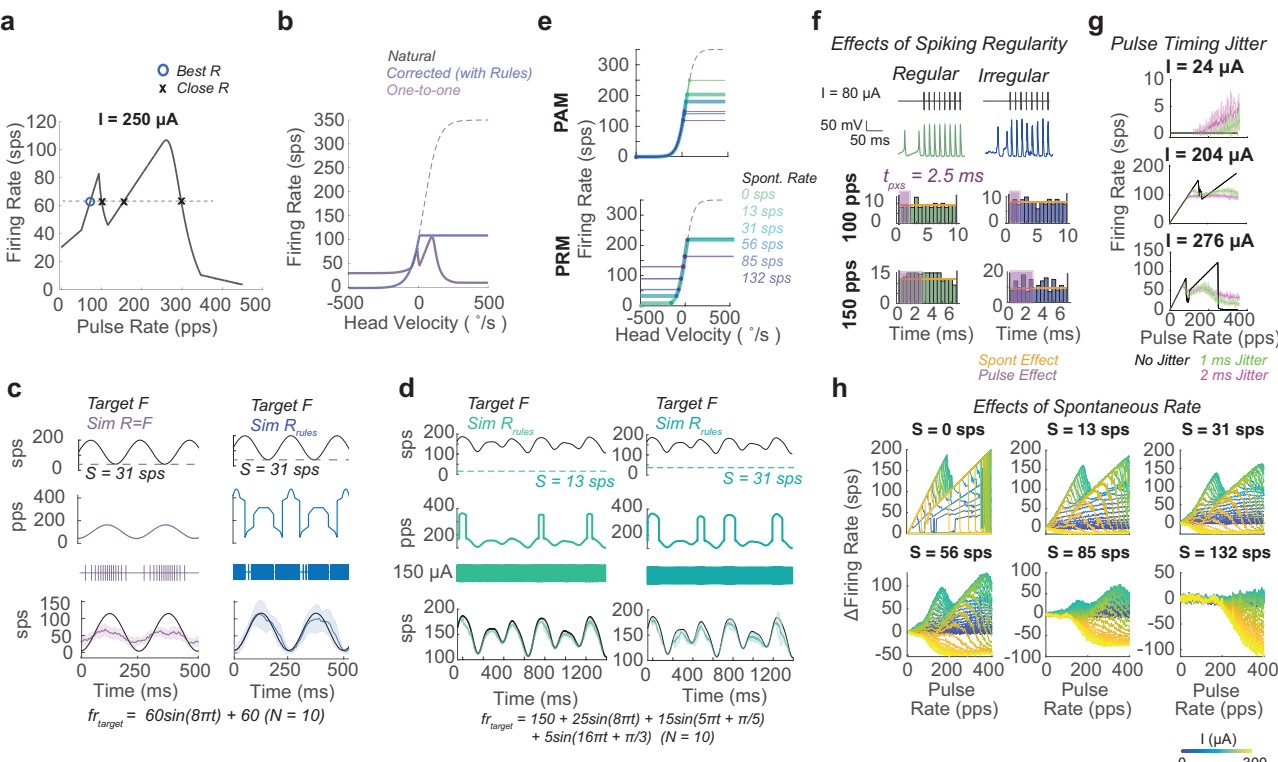

**Fig. 5 | Application of pulsatile effects to neural implants. a** Pulse rate-firing rate relationship at a high pulse amplitude standard for vestibular implants for an afferent with a spontaneous rate of 31 sps. Afferent used in **a**–**c**. Knowledge of this relationship can be used to predict optimal pulse rates for producing a desired firing rate. Here this is the lowest pulse rate that produces the desired firing rate (blue circle). **b** The natural head velocity to firing rate mapping (black dashed) compared to the standard one-to-one pulse rate mapping strategy (purple) and the corrected strategy (blue). **c** The target firing rate waveform (top), pulsatile waveform in pulse rate over time form and pulse sequence form (middle), and simulated responses waveform (bottom) for pulse rate modulation using the one-to-one mapping ($R = F$, purple) and the corrected strategy ($R_{rules}$, blue), using the equations from the paper to account for pulse effects. **d** The corrected strategy is

applied to afferents with two different spontaneous rates ($S = 13$ sps green, $S = 31$ sps teal) for target inputs that are a mixture of sinusoids. Displayed in same format as c. Data are plotted as mean ± SEM across simulations ($N = 10$) in bottom of **c** and **d**. **e** The head velocity to firing rate mapping for a pulse amplitude modulation and pulse rate modulation strategies across afferents with different spontaneous rates. **f** The timing of spontaneous action potentials between pulses for regularly (green) and irregularly spiking (blue) afferents. Shown as in Fig. 3B. **g** The PFR at different pulse amplitudes when pulse delivery timing has a no jitter (black), 1 ms (green), or 2 ms (pink) Gaussian noise in timing. **h** The response of simulated afferents of different spontaneous rates ($S$) replotted with panel per $S$ to emphasize the difference in the PFR at the same pulse amplitude, as amplitude increases (blue to yellow).

the graph reaches a range of significantly higher activation levels (Fig. 4g, see for statistics). These results reflect changes in the PFRs for $I < 70\,\mu A$ versus $250\,\mu A > I > 70\,\mu A$ in simulations (Fig. 4a). There were not enough afferents to test for spontaneous rate effects, but distributions are shown in Supplementary Fig. 6b.

Next, we assess how the pulsatile stimulation effects shown in this paper could alter the fidelity of desired firing patterns during standard stimulation paradigms. We take the case of vestibular prostheses where, standardly, the natural head velocity to firing rate mapping (black dash) is used to generate a target firing rate from detected motion; then, a one-to-one mapping between pulse rate and desired firing rate is used in a pulse rate modulation (PRM) strategy, under the assumption that at a high pulse amplitude, here 250 µA, each pulse will produce an AP[19] (Fig. 5a–c). With present stimulation algorithms, impaired vestibular ocular reflexes (VORs) are partially restored in the direction of increasing firing rate and less so in the direction of decreasing firing rate from baseline[29]. These results occur in afferents that have some residual spontaneous activity. We simulate this case in an afferent with spontaneous activity ($S = 31$ sps), receiving PRM to encode a sinusoidal eye velocity (Fig. 5a–c). The predicted head velocity to induced firing rate mapping can plotted by remapping based on the PFRs at these parameters. Using the one-to-one mapping (purple), the firing rate should not reach the maximum or minimum desired firing rate, and it shows a relative bias

towards being able to excite compared to inhibit (Fig. 5b). These responses reflect limitations in VOR observed in animals and humans with vestibular implants[29].

The equations described above can also be inverted to predict the optimal pulse rate -in this case the minimum pulse rate- for inducing a desired firing rate (see "Methods" section). Under the idealized assumption of similar neuronal activity across neurons, we see a monotonic encoding of head velocity can be restored using the same range of pulse amplitude and rate parameters (Fig. 5b, blue); it only requires a more complex but achievable modulation strategy (Fig. 5c–d blue). We then simulate the afferent response to each stimulation paradigm and see the predicted limitations in induced firing rate with the one-to-one mapping and the desired firing rate response from the corrected paradigm (Fig. 5c, d and Supplementary Fig. 7). Although the equations were derived from 1-s fixed-rate fixed-amplitude pulse trains, we see the rules explain the limitations of the one-to-one mapping and consistently predict stimulation patterns that can produce sinusoids and more complex mixtures of sines with high fidelity from individual afferents of various spontaneous rates with PRM and PAM paradigms (Fig. 5c, d and Supplementary Fig. 7c, d).

Our results indicate both potential improvements and limitations to using a pulsatile electrical stimulation paradigm. Accounting for pulsatile stimulation effects, stimulation paradigms could be modified to produce firing patterns closer to the desired patterns, and the

described mechanism of pulsatile stimulation indicates that those firing patterns will be achieved with high fidelity. Additionally, we find that these rules hold for afferents with regular and irregular spike timing but proceed at different rates with increased pulse amplitude (Fig. 5e, f and Supplementary Fig. 8). In the case of triggering errors, as simulated by jittering pulse timing by 1–2 ms, the rules also hold, but pulse effects are smoothed due to a similar effect to having on-going EPSC activity (Fig. 5g). At the same time, we find that pulse-spontaneous interactions create multiple sources of variability in induced firing rate. The range of inducible firing rates differs across afferents with different spontaneous rates (Fig. 5e and Supplementary Fig. 7), and afferents with different spontaneous rates or even different channel densities in the axon undergo different levels of additive and blocking effects in response to pulses of the same pulse amplitude (Fig. 5h and Supplementary Figs. 8 and 9). These sources of individual afferent variability lead to mixed effects on local population responses to pulsatile stimulation.

## Discussion

We use detailed biophysical models of vestibular afferents to investigate the sources of variability in producing desired firing patterns using pulsatile stimulation. Our simulations show a number of effects of pulsatile stimulation on axon channel dynamics that can prevent other pulses from producing APs and override spontaneous activity. The resulting PFRs resemble pulse effects demonstrated across neural systems: high-frequency facilitation (row 1) has been observed in auditory nerve fibers[27,30]; the PPB effect that leads to a bend in PFR (row 2) has been observed auditory nerve fibers[25] and dorsal column axons[22,27,30]; high amplitude block is observed in the sciatic nerve (row 3)[31]; amplitude-dependent growth of firing rates has been observed in the auditory nerve[32]; experiments on hippocampal neurons[33], auditory fibers[28] and spinal cord proprioceptive fibers[21] demonstrate pulse-spontaneous additive and blocking effects (Fig. 4a). These similarities further support our hypothesis that there is a large source of shared variability in effects of pulses in clinical applications due to pulses driving axonal channel dynamics to unnatural states. A positive outcome of this hypothesis is that producing algorithms that are capable of accounting for complex pulsatile interactions at the axon should be applicable across use cases and be able to improve a variety of neural implant algorithms.

In this paper, we demonstrate one way of transforming our understanding of pulse effects at the axon into equations. A beneficial attribute of present methods of electrical stimulation is that the regularity of fixed-parameter stimulation produces a consistent effect on the axon that can be fitted with computationally efficient, analytical equations. We show equations fitted to one-second blocks of fixed rate-fixed amplitude stimulation can predict responses to pulse rate and pulse amplitude modulation sequences and correct them for pulse effects with modulation on the 5-50 ms timescale (Fig. 5 and Supplementary Fig 7c, d). Corrections produce firing patterns in silico that under healthy neurological conditions could fully restore the VOR where previous parameterizations could not (Fig. 5). The modification of neural implant algorithms predicted with these equations can be tested with experiments in Fig. 5 for improvements in driving desired firing patterns.

In post-damage and implanted systems, lower levels of activity are expected, as in the implanted vestibular afferents in the data analyzed in this paper[9]; reduced responsiveness to stimulation may occur, as in explanted vestibular afferents compared to in vivo[25], or rate of spike-recovery may change as in post-deafness auditory nerve fiber under cochlear implant stimulation[34]. Whichever case, healthy and damaged neuron parameterizations could be made using the experiments in the text and with a measurement or estimate of spontaneous activity. Reduced responsiveness to stimulation could be captured in the parameterization of the pulse-spontaneous interaction parameters ($p_{sxp}$, $p_{p|s}$), and differences in temporal channel dynamics due to

damage or natural physiological differences in channels used to drive APs in other systems could be captured with adjustments of pulse-pulse parameters ($t_p$, $p_{pb}$, etc.).

Additionally, understanding the source of pulse effects, as we do for biphasic pulsatile stimulation here, may help to design novel stimulation waveforms with beneficial effects. For example, we show that the cathodic phase of pulses leads to the blocking effects, and the anodic recovery phase can affect the duration of the evoked spike afterhyperpolarization; using this information, the shape of the recovery phase of a pulse could be designed to sensitize the axon so that when the next pulse is delivered one-to-one AP induction occurs (Supplementary Fig. 10). We can use a similar analysis to that in the text to create equations that capture effects of these pulses.

Another important conclusion of this paper is that afferents with different spontaneous rates and even channel densities produce different levels of additive and blocking effects in responses to pulses of the sample pulse rate and pulse amplitude[35]. These results imply that our present uses of pulsatile stimulation are not producing coherent local excitation in most cases, due to the diversity of baseline neural activation levels. They produce a consistent but unnatural combination of local excitation and inhibition, where the response of a neuron is based on its ongoing level of activity and distance from the electrode site.

Our findings suggest several possible improvements to neural implants, even considering the mixed effects of pulses on neuronal populations with natural levels of diversity. A hardware solution that is already under development[35] would be to use high-density electrode devices and small amplitude stimuli that are capable of targeting individual neurons. Our study of pulse parameter effects also indicates a number of algorithmic improvements. One inference observable from low pulse amplitude simulations (i.e. Fig. 4a $I$ < 45 μA) is that the PFR would be highly linear for all spontaneous activity levels but with a low slope. Thus, a high-rate low-amplitude stimulation parameterization may induce nearly linear modulation that can induce the upper range of firing rates seen in the system (i.e. 1000 pps producing 500 sps). Additionally, using more complex optimization strategies to find parameters that best co-activate neurons with a range of spontaneous activity levels may be another useful way to use our equations. Still, characterizing a large number of densely packed neurons may be intractable presently, and, especially in highly interconnected areas, such as parts of cortex, time-varying inputs may be difficult to account for. Another potential solution indicated by our study would be to eliminate spontaneous activity or inputs from other areas. For example, one could use site-specific channel blockers or other neuronal silencing techniques[36]. This would make neurons easier to drive with consistency throughout the population because it eliminates pulse-spontaneous interactions and leads to a larger inducible firing range (Figs. 4a and 5h). Highly interconnected regions may remain difficult to characterize and isolate in this way.

These findings also raise questions about ongoing practices involving electrical stimulation. First, it calls into question whether electrical stimulation-based mapping studies unveil natural functional connectivity and behavioral relevance as opposed to some level of anatomical connectivity and functions of the most excitable local neurons. Additionally, despite these seemingly inconsistent and unnatural changes in population firing, the brain processes the stimulation-induced signals sufficiently to make significant clinical improvements. For example, cochlear implants effectively restore speech perception[37,38], although cochlear implant users[5,7] have remaining deficits like other types of implantees[5,7], such as lack of tone discrimination or the ability to hear speech-in-noise[39]. These outcomes suggest a potentially exciting direction for improving stimulation algorithms is to focus on neural signatures of coherent population-level encoding as opposed to producing a high-fidelity single-neuron response in targeted neurons in the population.

Using equations like those in this paper, or reduced forms of them, we can now begin to build larger-scale population models of the effects of pulsatile stimulation on local and interconnected populations performing functional tasks[40]. By exploring the effects of pulsatile stimulation in more realistic population models, we can not only improve our use of pulsatile stimulation but also gain insight into the unidentified and seemingly system-wide population-level computations in the brain that underlie successful pulsatile stimulation-based treatments today.

## Methods

### Biophysical modeling of vestibular afferents

Vestibular afferents were simulated using a biophysical model that has been used previously by several groups including our own to study the effects of electrical stimulation on vestibular afferents[24-26]. Past work from the lab showed this model can replicate experimental firing rates and changes in firing rate with pulsatile and direct current stimulation[25,26].

We use an adapted version of the Hight and Kalluri model[24-26]. In brief, Hight & Kalluri showed that vestibular firing can be simulated accurately by assuming cells have the same shape and size. Type I and Type II vestibular afferents are modeled as differing only in channel expression and EPSC magnitude ($K$). Spontaneous rate can be set by changing the average inter-EPSC arrival interval ($\mu$).

The membrane potential ($V$) varies as:

$$\frac{dV}{dt} = \frac{1}{(C_m S)}(-I_{Na} - I_{KL} - I_{KH} - I_{leak} + I_{epsc} + I_{stim}) \quad (5)$$

where in addition to the current from each channel type, the membrane potential is influenced by the EPSCs arriving at the axon ($I_{epsc}$) and the injected current ($I_{stim}$). The overall current through the membrane in the denominator is dependent on individual membrane voltage-gated channel conductances: Na ($g_{Na}$, m, h), KH ($g_{KH}$, n, p), KL ($g_{KL}$, w, z). We simulate the electrode at 2 mm from the simulated afferent which causes the firing threshold to be around 56 $\mu$A for a typical neuron.

For this study, we adjust simulation parameters to reflect the irregularity and baseline firing rate of a vestibular afferent recorded in previously published findings[9]. Out of the dataset, we fit the afferent that showed the largest diversity in PFR functions in response to pulses of different pulse amplitudes, expecting it would show the most diversity in pulse effects. We find that conductance values of $g_{Na}$ = 13 mS/cm$^2$, $g_{KH}$ = 2.8 mS/cm$^2$, and $g_{KL}$ = 1 mS/cm$^2$ and EPSCs with $K$ = 1 and $m$ = 1.3 ms match previously published experimental findings at pulse rates from 25 to 300 pps. We keep these conductance values for all irregular afferent simulations in the main body of the text.

For studies of the effects of spontaneous rates on firing, the channel conductance values are kept the same but the inter-EPSC arrival interval $\mu$ is set to 0.25, 0.5, 1, 2, 4, and 8. To model the axon with no spontaneous activity, $I_{epsc}$ was set to 0.

Additionally, we assess the effect of firing regularity on induced firing rate. The irregular neuron ($F$ = 36.6 ± 0.9 sps, $CV$ = 0.57, where $CV$ is the Coefficient of Variance), is modeled with $K$ = 1, and $\mu$ = 1.65 ms. A conductance matched regular neuron ($F$ = 33.8 ± 0.4 sps, $CV$ = 0.09) is also modeled with $g_{Na}$ = 13 mS/cm$^2$, $g_{KH}$ = 2.8 mS/cm$^2$, and $g_{KL}$ = 0 mS/cm$^2$, $K$ = 0.025, and $\mu$ = 0.09 ms.

The effects of channel conductance values on the PFR are tested while repeating the sample pulse block experiments. We use a biologically realistic case by using lower conductance values and changing parameters to produce firing rates and regularities similar to those observed in a previous in vitro experiment with and without exposure to direct current[41]: $g_{Na}$ = 7.8 mS/cm$^2$, $g_{KH}$ = 11.2 mS/cm$^2$, and $g_{KL}$ = 1.1 mS/cm$^2$, $K$ = 1. $\mu$ was again varied from 0.25 to 8 ms.

We find no evidence of pulsatile stimulation affecting the hair cell, so all direct current-related hair cell effects (adaptation, the non-quantal effect, etc.) are not activated in these simulations[25]. The simulation is run using the Euler method to update all variables through each of the channels.

### Simulated pulsatile stimulation experiments

We replicate the experiments from Mitchell et al.[9] in silico with a finer sampling of pulse amplitudes and pulse rates. In addition to the pulse rates used experimentally, pulse rates from 1 to 350 pps in steps of 1 pps are delivered for 1 second. Ten repetitions are performed for each current amplitude, spontaneous rate, and pulse rate combination. Pulse amplitude is varied from 0 to 360 $\mu$A in steps of approximately 12 $\mu$A and used to parameterize equations values. Around transitions in the level of pulse effects, PFRs are simulated in finer detail to capture the change in effect sizes. We interpolated between these values to create a smooth function for predicting induced firing rates.

This combination of experiments is repeated on the irregular neuron, regular neuron, and low conduction/in vitro neuron. It is also repeated for all values of $\mu$ to map how pulse effects change with different levels of spontaneous activity.

### Jitter experiment

To assess the effect of jittered pulse delivery time on induced firing rate, we perform the same simulation but include jitter in pulse timing. Instead of delivering perfectly timed pulses, we add a Gaussian noise term with a standard deviation of 1 ms or 2 ms to the exact pulse timing to simulate delay or advancement in the delivery of regularly scheduled pulses.

### Pulse rate and amplitude modulation

To test how the pulse rules apply to sinusoidal modulation, as used in various prosthetic algorithms, PRM and PAM were simulated with pulse parameters restricted to the range commonly used in vestibular prostheses: pulse amplitudes 0 to 350 $\mu$A and pulse rates between 0 and 360 pps[5,19,42]. We use a simple optimization strategy, as a demonstration of the applicability of these equations. For PRM, the common vestibular prosthetic strategy, a PFR is generated at the chosen pulse amplitude based on the equations. Then, the lowest pulse rate that produces the target firing rate desired (or the closed firing rate achievable using rms) is selected (Fig. 5a). For PAM, in an analogous manner, the chosen pulse rate is selected, and the pulse amplitude-firing rate mapping is used to select the lowest pulse amplitude that produces the desired firing rate (or the closed firing rate achievable using rms). Potential pulse amplitudes and rates were sampled in steps of 1 $\mu$A and 1 pps. This solution was a simple approach for minimizing energy consumption in either stimulation paradigm. For a moving firing rate prediction in the text, the target firing rate trajectory is sampled at 0.1 ms sampling frequency, and optimal pulse parameters are chosen at each time step.

In the main text, we simulate a vestibular afferent with a low level of residual activity, $S$ = 31 sps responding to PRM encoding a pure sinusoid with a fixed-amplitude of 250 $\mu$A. We use a standard one-to-one mapping strategy, as used in vestibular prostheses[16,19], transforming head velocity into the desired firing rate into the delivered pulse rate. We compare to the optimal PRM using the strategy described above. In Supplementary Figs., we also use afferents of different spontaneous rates and target firing patterns that are mixtures of sinusoids within the range of velocities experienced by the human vestibular system, less than 10 Hz. We use PRM sequences with a fixed amplitude of 150 $\mu$A, the largest firing range observed as pulse rates vary. In the PAM cases, the pulse rate is fixed at 100 pps, the maximum observed firing rate for PAM, while pulse amplitudes are varied. We assess the PAM optimization described above.

## Experimental data

We reanalyze 6 afferents recorded from rhesus monkeys during the experiments for Mitchell et al.[9]. All procedures were approved by both the McGill University Animal Care Committee and the Johns Hopkins Animal Care and Use Committee, in addition to following the guidelines of the Canadian Council on Animal Care and the National Institutes of Health. See Mitchell et al.[9] for experimental details and spike-sorting information. For each afferent, the current amplitude that produced facial twitch was found. The 100% pulse amplitude was 80% of the level that produced facial twitch for a given electrode site. PFRs are sampled at 25–300 pps in steps at 25%, 50%, 75%, 87.5%, and 100% pulse amplitude. For some afferents, only 100% amplitude trials were recorded. There were 3-5 repetitions per pulse amplitude. Sorted spike times were used to calculate the average firing rate during each block. Pulse delivery times were analyzed in the equivalent way to get the pulse rate per session. The firing rate between blocks was used as the spontaneous rate per afferent.

## Comparison of experimental and simulated afferent responses

Slopes between sequential pulse rate-firing rate experiments are calculated for each experimentally recorded afferent, revealing a trend of change in slope as the pulse rate increases. A similar analysis is performed on simulated afferents. The simulated experiments are sampled every 30 pps to compare the PFRs. The slopes are sampled every 60 pps. The pdfs of the simulated and experimental slopes are compared. We expect peaks at $1/n$ and changes in the distribution of slopes as pulse amplitudes increase. A Welch $t$ test, Wasserman distance, and Kolmogorov-Smirnov test are used to compare the similarity between distributions. Significance is also assessed with a permutation test. The matching of experimental firing rates to experimental pulse rates is shuffled across afferents. Then, the slopes per simulated afferent set are recalculated 5000 times to find the probability of these statistics occurring by chance. Afferents are grouped by spontaneous rate closest to simulated levels into 3 groups. The same comparisons are made with simulated data at these same pulse rates (Supplementary Fig. 5b).

Experimental data is further compared to itself in two ways. The PFR is split into two sections: low versus high pulse rate ($R$) at $R = 150$ pps. Slope distributions are compared under these conditions with a Welch t-test at the maximum $I$ tested per recording ($n = 9$), expecting significant difference due to simulations above $I = 75$ μA, showing strong PPB and SFP. Additionally, effects of pulse amplitude are assessed, by comparing the low R versus high R PFRs at low $I$ ($I < 0.5I_{max}$) versus high $I$. This comparison was made for slopes and normalized area under the curve (AUC) of the PFR:

$$\widehat{AUC} = \sum_{i=1}^{N-1} \frac{(F_i + F_{i+1})(R_{i+1} - R_i)}{2(R_{i+1} - R_i)} \tag{6}$$

A Welch $t$-test was used for comparisons of these parameters, as well.

## Parameterizing fits

The optimal parameterization of the equations is found using *patternsearch* in Matlab in the "classic" generalized pattern search algorithm mode which requires parameter initializations and the bounds to be set for each parameter. For a subset of fourteen of the PFRs at simulated pulse amplitudes, the starting parameterizations were found by hand for each of the spontaneous rate cases. At $S = 0$ and $S = 56$, three additional $I$s were sampled, focusing on the transition points to capture the rule transitions accurately ($I \in [30-100]$ and $I \in [150-250]$). The maximum and minimum $I$ cases were included in this group. These fits are referred to as hand-fitted. For the remaining $I$s, linear interpolation between the fitted $I$s followed by optimization is used to obtain optimal parameters. This technique was done to increase the chance of optimization finding solutions involving smooth changes in parameter values that reflect the observed mechanism of AP generation. For fitting details of the parameters, see Supplementary Table 1, and for observation of the parameterization across $I$ and $S$ conditions, see Supplementary Fig. 2.

Standard rms error is used for optimizing the best fit at each amplitude. Data are fit to the mean of across simulations. The fit is reported for error across each of the ten simulated runs per model. The difference between error levels of fitted and interpolated PFRs is assessed with a paired $t$-test. Data is all reported as mean rms across repetitions ± sem.

A sensitivity analysis was performed on the optimized parameterization of the fitted $I$ cases. All optimized parameters were held fixed except for one which was jittered 100 times within a Gaussian range of 10% of the optimized value. The effect on rms between predicted and simulated PFR was then assessed and reported in Supplementary Fig. 4.

## Predictive equation

The observed effects at the axon are transformed into equations that depend on measurable or controllable variables: pulse amplitude ($I$) delivered from the electrode, pulse rate ($R$), and spontaneous rate ($S$). We find that due to the fixed pulse rate and pulse amplitude of a train of pulses, we can transform the effect of pulses on the complex system of channels driving an action potential into equations that capture the observed effects of pulses and their mechanisms without dependence on time.

We find that $I$ has the strongest effect on the type of interactions pulses have, but $S$ also affects most of the parameters of that control pulse effects. All variables that depend on $I$ and $S$ are bold below and written the first time with their dependencies. In most cases, $S$ acts to either deepen the blocking or shallow the additive effect of pulses.

**Pulse-pulse interaction equations.** The main pulse-pulse effects rely on pulses having refractory effects on channel opening and closing that will affect following pulses.

A pulse-pulse blocking effect begins with a period in which another pulse would be blocked with 100% certainty $t_b(I,S)$. If the interpulse interval ($1/R$) is longer than that window, then no blocking occurs. If the pulse rate is high enough, more pulses fall into $t_b$ and are blocked. We model the resulting firing rate with a ceil function:

$$F_{pp} = \frac{R}{R/R_b}, \text{ where } R_b = 1/t_b \tag{7}$$

Instead of changing from producing one AP to every other pulse making an AP, there is an intermediate set of $Rs$ starting at $R_{pb}$ at which sequences of pulses build towards a block effect for a pulse in a sequence of at least three pulses. This effect resembles facilitation but where the result is a pulse being blocked instead of producing an AP. This effect always begins at a lower $R$ than $R_b$, so it can be parameterized in a bounded way as a fraction of $R_b$, which simplifies optimization:

$$R_{pb}^n = (n - \boldsymbol{p_{pb}^n})R_b \tag{8}$$

For an intuitive picture, we can say that, on average, the axon experiences an extended refractory period in which the probability of a pulse being blocked by a previous pulse is 1 until $t_b$ ms after the pulse and returns to zero when the inter-pulse interval exceeds $t_{pb} = \frac{1}{R_{pb}^n}$. We observe that partial block at $R_{pb} < R < R_b$, which on average is like time $t_b < t < t_{pb}$, where $t = 1/R$. Under most conditions, the partial block period takes a simple form, transitioning from 0 to 1 with increasing $R$, but under special cases, the window extends as the pulse creates harmonics in the voltage-gated channels opening and closing. In all

cases, it can be defined as an added probability of a pulse getting eliminated, $\psi$.

$$F_{pp} = \frac{R}{\lceil R/R_b \rceil + \psi(I, S, R)} \quad (9)$$

$\psi$ is the sum of the effects of each of the $n$ bends:

$$\psi(I, S, R) = \sum_{n=1}^{N} \psi_n$$

In most cases, $\psi_n$ takes the form of a linear increase in the probability of a pulse surviving the refractory period as the time of initiation is closer to $t_b^n$ than $t_b^n$. This effect scales with $k_{pb}^n(I, S)$

$$\psi_n = \begin{cases} \min & \left\{ 1, (1 + \kappa_{pb}^n(I, S)) \left( 1 - \frac{\frac{1}{R} - \frac{t_b}{n}}{(t_{pb}^n - \frac{t_b}{n})} \right) \right\}, t_{pb}^n < \frac{1}{R} < \frac{t_b}{n} \\ 0 & , else \end{cases} \quad (10)$$

We made the simplifying assumption that for the 2nd to $n$th bend, all refractory effects are the same. So, we only parameterize $p_{pb}^1$, $p_{pb}^2$, $k_{pb}^1$, and $k_{pb}^2$, where the second, $p_{pb}^2$ tends to be larger.

**Suppression of future pulses.** When the pulse amplitude is sufficiently high, in these simulations >180 μA, pulses can prevent all firing for extended periods of time. So, for the $R$ after which the second pulse would be blocked a very steep refractory effect with scaling $\kappa_{pb}(I, S)$ occurs:

$$for \, \frac{1}{R} \geq t_{pb}^2,$$

$$\psi_2' = \kappa_{pb}^2 \left( 1 - \frac{\frac{1}{R} - \frac{t_b}{2}}{(t_{pb}^2 - \frac{t_b}{2})} \right)^3 \quad (11)$$

In this same range of pulse amplitudes, there is a small window in which pulses can prevent firing for a number of pulses, as channels enter an extended dynamic loop, but, once $R$ exceeds $1/t_{pb}^1$, the standard $F = R/2$ PFR is restored, until $1/R > t_{pb}^2$.

**Pulse dynamic loop (PDL).**

$$for \, t_{pb}^1 < 1/R < t_b,$$

$$\psi_1' = \left\lceil \kappa_{pb}^1 \left( \frac{1/R - t_b}{t_{pb}^1 - t_b} \right) \right\rceil \quad (12)$$

Then,

$$F_{pp} = \frac{R}{\min\{2, \lceil t_b R \rceil\} + \psi_1 + \psi_2} \quad (13)$$

An additional special case is at low pulse amplitudes and when $S = 0$. Pulses only facilitate themselves at sufficiently high pulse rates. This was represented as a sigmoid function with a scaling, slope, and center that depended on $I$:

$$F = \frac{1}{(1 + \exp(m_{facil}(R + R_{ppfacil})))} F_{pp} \quad (14)$$

**Pulse-spontaneous interaction equations.** Spontaneous activity has additional blocking and facilitation effects in the presence of pulses as described in the term $F_{ps}$. There is a probability that a pulse can make an action potential given the presence of enough EPSC activity to produce a given spontaneous rate $S$, $p_{p|s}$. We model these effects

proportional to $S$. So, the final firing rate will be a combination of the natural activity $S$, the contribution of pulse-spontaneous interaction effects, and the contribution of pulses to firing, scaled by this resistance to pulse effects $p_{p|s}$:

$$F = \max\{0, p_{p|s} F_{pp} + F_{ps} + S\} \quad (15)$$

Then, there are two linearly increasing blocking effects. First, spontaneous activity blocks pulses at a set level because it is approximately normally distributed between pulses (Fig. 3b yellow). Conversely, pulses block a linearly increasing number of spontaneous action potentials, as the pulse rate increases, and create a longer and longer window $t_{pxs}$ after a pulse in which all EPSCs are blocked from becoming action potentials (Fig. 3b purple). This results in two linear functions of $R$. One difference is that the blocking of spontaneous activity by pulses is not ever present, so there is some pulse rate, $R_{pxs}$ after which the pulses can block spontaneous action potentials. This value decreases to 0 as current amplitude increases and all pulses can disrupt natural action potentials.

$$F_{ps} = p_{psfacil} R + \max\{-S p_{p|s}, -p_{p|s} p_{sxp} R\} + \max\{-S, p_{pxs}(R - R_{pxs})\} \quad (16)$$

Finally, at low current amplitudes, facilitation occurs between pulses and spontaneous activity. This effect is most easily observed in the range when the afferents with no spontaneous activity show no firing. Facilitation ends when $p_{p|s}$ approaches 1, as pulses transition from needing facilitation to being sufficient to produce action potentials to producing refractory effects that create pulse-spontaneous blocking.

These equations can also be rewritten to estimate the contribution of pulses versus spontaneous activity to the final PFR *as:*

$$F_P = p_{p|s} F_{pp} + p_{psfacil} R + \max\{-S p_{p|s}, -p_{p|s} p_{sxp} R\} \quad (17)$$

$$F_S = S + \max\{-S, p_{pxs}(R - R_{pxs})\}$$

### Reporting summary
Further information on research design is available in the Nature Portfolio Reporting Summary linked to this article.

## Data availability
The datasets generated in this study have been deposited in the Zenodo database (https://doi.org/10.5281/zenodo.11387800) and within the provided Source Data file. Source data are provided with this paper.

## Code availability
The custom code used to generate simulations, implement the pulse prediction rules, and fit the data during the current study are available from the corresponding author on request. A version of the code is available on https://github.com/CSteinhardt153/pulsatile-prediction-code.

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

## Acknowledgements

The authors would like to thank the Simons Society of Fellows 965377, Gatsby Charitable Trust GAT3708, and the Kavli Foundation for supporting C.R.S., NIH R01NS110893 and R01DC018300 for supporting G.Y.F., and NINDS R01DC002390 and NIH R01DC018061 for supporting K.E.C. and D.E.M. during this work.

## Author contributions

C.R.S. performed computational simulations, data analyses, and equation development. C.R.S. and G.Y.F. developed the investigation

question. D.M. performed experimental studies. C.R.S., G.Y.F., and K.E.C. produced the manuscript. G.Y.F. and K.E.C. supervised the work.

## Competing interests

The custom prediction code in this manuscript is part of PTC patent WO2022178316A1 filed by Johns Hopkins University and a national filing at this time with inventors C.R.S. and G.Y.F. The remaining authors declare no competing interests.
