## [Peer Review File · Nature Communications]

Pulsatile electrical stimulation creates predictable, correctable disruptions in neural firingREVIEWER COMMENTS

Reviewer #1 (Remarks to the Author):

This manuscript describes a computational modelling study, investigating the temporal interactions that occur in a model vestibular nerve fiber due to the behavior of the model's voltage-gated ion channels in response to both electrical stimulation from a neural prosthesis and spontaneous synaptic inputs from a vestibular hair cell. Simulation results are compared to published electrophysiological data from vestibular nerve fibers in rhesus monkeys. The goal of the study is to not only characterize the pulse-pulse and pulse-spontaneous temporal interactions, but also to develop quantitative descriptions of these interactions that can be used to develop optimized stimulation strategies for neural prostheses.

Overall, the study is well conducted and the manuscript well written. The authors have carried out a very thorough analysis of the model's response behaviors and an excellent comparison with the published electrophysiological data. I do have some concerns that the authors have somewhat overstated the novelty and clinical applicability of this work.

Regarding the novelty, previous modeling studies of the interaction between electrical stimulation and synaptic input have been conducted, particularly in the context of DBS (see the recent review by Ng et al. in *Neuromodulation* - <https://doi.org/10.1016/j.neurom.2023.04.471> - for a nice summary) and cochlear implants (see Kipping & Nogueira *JARO* 2022 - <https://doi.org/10.1007/s10162-022-00870-2>). A more thorough literature review to place the present study in the context of the articles that I have mentioned and other related publications would be beneficial.

With respect to the clinical applicability, the model and the physiological data are describing electrical stimulation from a vestibular prosthesis that is implanted in a healthy animal. There is an extensive literature on how short-term and long-term deafness leads to changes in the response of auditory nerve fibers to stimulation from a cochlear implant. I am not aware of similar studies in the case of vestibular pathology, but it is likely that analogous changes occur in vestibular nerve fiber responses due to damage to vestibular hair cells and subsequent changes in the connected nerve fibers. In addition, as the authors point out, in a population of nerve fibers receiving independent synaptic input, the ability to pattern the electrical stimulation to optimize responses might be greatly limited by the great range of patterns of synaptic input those fibers receive, and for a clinical vestibular prosthesis, there may be no practical way of measuring what synaptic input the population of fibers is receiving. These issues need to be discussed in the manuscript.

Finally, the main equation describing the model, Eq. 5 in line 517 of the manuscript, is incorrect. All of the currents on the righthand side of the equation should be in the numerator, not the denominator, and the voltage-gated ion channel currents and the synaptic current should all have a minus sign in front of them. I believe this is a typesetting error, rather than being an error with the model implementation itself, since the model behavior is as expected. I think that this typesetting error has occurred in transcribing the corresponding equation from the authors' previous modeling article (Steinhardt & Fridman, 2021). In that previous paper, the membrane equation is given correctly on page 2 of the supplemental material, but on page 1 in Eq. 1 of the supplemental material, the minus signs are missing, and the equation is typeset ambiguously using the notation $1/(CmS)(INa + IKL + IKH + I_{leak} + I_{epsc} + I_{stim})$, where the 1/ only corresponds to the first term in parentheses, but in the manuscript under review the second term with the currents has erroneously been placed in the denominator as well. This needs to be corrected.

Reviewer #2 (Remarks to the Author):

I read this paper with interest, the authors describe the effects of “pulsatile” stimulation on vestibular axons and try to describe effects that occur at axonal level and that impact the neural output against the naively assumption of $F=R$. Using computational biophysics, they study the complex effects that affect firing rates for different rates and pulse amplitudes with and without the presence of spontaneous neural activity and then they show that recordings of single vestibular fibers do show some of the properties they found. While I think this work is very interesting and timely, I think it fails to provide clear message and example of what this means for stimulation systems and how we should change the way we designed them other than “increase the number of contacts”. Therefore, I think the paper requires some re-thinking to be able to nail down the real impact of the results.

Below my comments.

Novelty and literature

While I understand that the authors are focused on vestibular stimulation and therefore are knowledgeable of the literature in that field. Since they pitch the paper as to studying fundamental principles that apply “in general” in neurostimulation they have to know that these pulse-pulse and pulse-spontaneous activity interactions has been studied already in other applications. Specifically I would suggest them to look at the papers of Formento et al Nature Neuroscience 2018 and the preprint from Steve Prescott lab (<https://doi.org/10.1101/2023.01.10.523167>). None of this removes importance from their findings, but I think they must be put in context. For example, the explanation of the pulse-pulse interaction is essentially describing pulse skipping, or the pulse-spontaneous activity, is describing cancellation of ongoing natural activity, with relationships that have been previously observed and explored in computational models.

Figures

The figures are difficult to understand, and legends do not help. Indeed, they aren't legends, but further description of the results. The legends should describe the elements represented so that one can understand what's being represented.

Validity and generalization of the mathematical model

The authors present their model capturing general principles to describe the effects on Firing rates of the different phenomena that can occur at the axon level. While I agree that these can be described with equations, that "model" does remain a fit to data. This means that, obviously, the equations are only valid for the specific set of simulated axons. As the authors point out indeed, different ion channel distribution affects all these phenomena, so their "model" would have to be refitted to the specific models of other neurons in order to generalize to different applications. For example, it would not hold for spinal motoneurons that have very slow calcium dynamics.

Relevance for neurotechnology applications

While I really think that this relationship must be explored the paper falls short at providing a clear example of how their model is going to help. Specifically, the authors report in Figure 5 that their equations can be used to produce more accurate sinusoidal firing rates. However, nowhere in the paper is shown what exact stimulation protocol is the model suggesting. This is very important because the output could be unfeasible. The authors described it as "still feasible" but provide no data to support that statement. The only additional figure is supplementary figure 7 that shows the effects of different pulse shapes one of which, the "asymmetric pulse shape", it's what's being used in neurostimulators from Medtronic since the 70s. So how exactly is the model changing the way we stimulate, if that's the goal?

An additional problem with the relevance is whether there is a problem at all. I am a fan of general theoretical papers, but the authors failed to identify a specific flaw in current applications. For example, cochlear implants work well, as they describe, even if the firing rates are all affected by the pulse-pulse problems. I think it is important to pinpoint a specific application in which the fact that F is not R does affect clinical outcomes, then, in that case, it will become very clear why this model helps.

All these points could have been addressed in the Discussion section, instead, the discussion is short and broad. The topics addressed in this work have significant implications and I would have enjoyed a detailed discussion. For example, in the discussion one of the consequence of their work, according to the authors is to increase the number of stimulation channels. But this "solution" is the same that is being proposed since the start of neuroengineering. So, what is the single important impact that knowing how the pulse rate is affected changes future and present solutions? Or could change them? This is not clear right now and the conclusion seems to be only the old obvious ones.

In conclusion I think the authors are on the right path and their work could be of big impact but it requires a significant revision of the current form to be able to impact neurotechnology practice.

Minor comments:

Fig. 2: It would be informative to include the sodium ion dynamics during spontaneous activity and natural channel perturbations.

Fig. 2e: For clarity purposes, it would be helpful to show an entire interpulse interval that indicates t_b and t_{pb} .

Questions:

Line 313: shapes like Fig. 3b left? I do not understand what shapes I should be looking at.

Line 330: reversal of effects? I do not see the reversal of facilitatory effects in the figure.

Line 335: cathodic block? First, time mentioned on the text as distinguishing the anodic and cathodic. I believe it requires an introduction

Line 350: 12 current amplitudes?

Fig. 4b: why variability in low S ? I would have expected well-defined behaviours

Line 381: 0.5-1 slopes PDF? Why not 0-1?

Line 394: simulations of head velocity? This information seems to not be provided.

Fig. 5g what is showing that has not been shown earlier?

Reviewer #3 (Remarks to the Author):

In this manuscript, the authors describe the use of computational models to explain the variability of outcomes normally observed with electrical stimulation of peripheral nerves, and specifically to vestibular afferents.

I recognise the importance of the development of simplified yet rich descriptions of neural response such as the one presented in this work. In particular, mechanisms of response to high-frequency stimulation are of particular interest, and in this manuscript are well described both mathematically and physiologically, in terms of ion channel dynamics. However, there are some important issues of that should be addressed.

Since this is a modelling study, the authors must publicly release the source code implementing the designed equations, with examples of its use, source data and well-detailed instructions for use. It is mandatory both for the results validity, and since as it could be a valuable tool for other researchers. This can be done in public repository of authors' own choice.

The model seems to fit the reference data produced by experimental and computational modeling very well. However, given the great freedom in the choice of parameters (mainly, in the manual definition of the various ψ_n and all the “p”s in the spontaneous-pulse interactions), this is not surprising. Therefore, sensitivity analysis of model outcomes w.r.t. these parameters should be performed: this means shuffling parameters in % range of their nominal values, and studying the outcomes stability.

Regarding validation against experimental data, the arguments are quite limited. The choice of sparsely sampling the simulated data, which yields non-significance the in KS test, is hardly justifiable. Of course instead there is significant difference when compared to 5000 permutations, it is not a fair comparison. In general, I recommend to report the reference experimental data from Mitchell et al. more thoroughly and to try to perform qualitative or quantitative comparisons in a less convoluted way. What is currently reported in Fig. 4d and used for validation seems quite reductive compared to the experimental dataset. Additionally, the authors could refer to the works they cite in the beginning of Discussion (lines 460-461) or similar studies from neighbouring fields in neurostimulation (there are successful validations cases in PNS neurostimulation and modelling studies).

The authors assert that they invert the equations to predict the PFR and then to find optimal parameters. Is this proper analytical inversion or are we talking about numerical solutions?

In Fig. 1c, it is not clear if the pulse current of 230 μ A used in the simulation corresponds to the experimental one. If not, reasoning should be provided.

In Fig. 2. Are the plots in the bottom row meant to be there? They are not referred to in the caption.

It is not clear what is represented on the right in Fig.3b , is it the distributions of distances in time between spontaneous APs and the previous pulse? It should be made clearer in the caption.

Also in Fig. 3b, why aren't $p_{psfacil}$ and p_{sxp} represented? It could help comprehension. In fact, the form of p_{sxp} is never explicitly reported in the manuscript.

In Fig. 3d are reported results for S between 0 and 132 sps, but in the legend is only listed 13 sps.

In general, a legend or table of used symbols and their explanation would aid the reader, who instead has to continuously fish through the document. Also, it would be interesting to see reported all parameters in function of S, I, R... as was partly done in Fig. 3b.

The equations should be recalled properly between results and methods: E.g., 3 and 16. It can be confusing especially if they are formulated differently.

Lines 331 to 334 are unclear and a parallel with Fig. 3d middle is not evident. The bend does not seem to appear at R_knee. Also, because p_sxp in function of S is never reported, it is unclear how R_knee should behave.

Also, in Fig. 3d middle, it is not clear why $\Delta F = S$, should it be placed on Fig. 3d right with a negative sign?

Regarding the testing of accuracy (lines 349-257), it is not clear how and where these 10 amplitudes for parametrization and 10 test amplitudes were selected, among the ~30 steps between 0 and 360 uA? Related to this and Fig. 4b, it is not clear on what pairs the paired t-test was applied.

Line 360, The count of afferents is not clear, six afferents, 5 and 4...?

In Fig. 4d it is unclear what is represented in the right column with respect to the left column.

Much more convincing is the perspective application of the identified equations in the optimization of stimulation paradigms to obtain desired firing rates, which has considerable practical implications.

Minor comments:

Line 31, applications.

Line 92, “spontaneous evoked” seems an oxymoron. I would see it as “evoked” unless argued differently.

Line 111, to match

Line 219, it is unclear what the term “bend” exactly refers to, the points of decrease in FR during PR increase?

Reviewer #1 (Remarks to the Author):

This manuscript describes a computational modelling study, investigating the temporal interactions that occur in a model vestibular nerve fiber due to the behavior of the model's voltage-gated ion channels in response to both electrical stimulation from a neural prosthesis and spontaneous synaptic inputs from a vestibular hair cell. Simulation results are compared to published electrophysiological data from vestibular nerve fibers in rhesus monkeys. The goal of the study is to not only characterize the pulse-pulse and pulse-spontaneous temporal interactions, but also to develop quantitative descriptions of these interactions that can be used to develop optimized stimulation strategies for neural prostheses.

1. Overall, the study is well conducted and the manuscript well written. The authors have carried out a very thorough analysis of the model's response behaviors and an excellent comparison with the published electrophysiological data.

We thank the reviewer for acknowledging that the simulations and experimental comparisons are well conducted to support our claims.

2. I do have some concerns that the authors have somewhat overstated the novelty and clinical applicability of this work. Regarding the novelty, previous modeling studies of the interaction between electrical stimulation and synaptic input have been conducted, particularly in the context of DBS (see the recent review by Ng et al. in *Neuromodulation* - <https://doi.org/10.1016/j.neurom.2023.04.471> - for a nice summary) and cochlear implants (see Kipping & Nogueira *JARO* 2022 - <https://doi.org/10.1007/s10162-022-00870-2>). A more thorough literature review to place the present study in the context of the articles that I have mentioned and other related publications would be beneficial.

We thank the reviewer for bringing to our attention that the novelty and clinical applicability of the work in the context of previous studies were not clear to the reader in the previous version of the manuscript.

To address the novelty of our work, we have added a more thorough literature review of models of electrical stimulation of neurons across a number of neural implant applications to the Introduction. In this context, we highlight the novelty of our work in focusing on developing a general method that advances the ability of researchers and clinicians to control firing rate over time. We now include the DBS work mentioned above as well as work from other systems in our revised Introduction, specifically: *“Neural engineers have explored the factors that impair neural implant performance using detailed biophysical models that include neuron-specific channels, ion densities and physiology¹²;.... Particularly, the deep brain stimulation (DBS) field has used this approach to understand the impact of parameters such as pulse waveform, electrode orientation, and tissue properties on neural activation¹³. Successes in this field have led to the use of patient-specific modeling as a popular clinical approach for finding patient-specific stimulation parameters that improve the performance of a variety of implants^{11,14,15}. These parameterizations however do not account for another essential feature of neural responses: the neuronal firing pattern over time..... Here, we take a different approach to this question: we use a detailed biophysical model to investigate factors of spontaneous activity and pulse parameterization that impact firing rate and extract general principles of pulsatile interactions from the simulation. We use these rules to generate time-independent equations that can estimate the induced firing rate in response to pulse parameters and could be parameterized for various neuronal systems based on measurable observations of the system (p.1-2)*

We also add a discussion of previously observed pulse-spontaneous interactions across systems and attempts to create simplified equations for estimating pulse effects(in the new Results section), including works such as the cochlear implant paper mentioned above: *“Prior work by our group and others has attempted to capture these interactions using simplifying equations^{26,28}, but those attempts do not provide a complete description of the effects observed in our simulation described below.” (p.9)*

To address clinical applicability, we now address the potential use of these equations in neural implant algorithms in our revised Introduction:

“An advantage of this approach is that resulting equations can be inverted and integrated into real-time devices to correct for complex effects of pulses on firing rate in a computationally efficient way, improving our ability to precisely control neural firing rate over time.” (p.2)

Finally, we have added several new paragraphs to our revised Discussion that specifically address how to tune the equations to different systems and overcome some challenges of using them in various neurological systems:

“In this paper, we demonstrate one way of transforming our understanding of pulse effects at the axon into equations. A beneficial attribute of present methods of electrical stimulation is that the regularity of fixed-parameter stimulation produces a consistent effect on the axon that can be fitted with computationally efficient, analytical equations. We show equations fitted to one-second blocks of fixed rate-fixed amplitude stimulation can predict responses to pulse rate and pulse amplitude modulation sequences and correct them for pulse effects with modulation on the 5-50 ms timescale(Fig. 5, Supp. Fig 7c-d). Corrections produce firing patterns in silico that under healthy neurological conditions could fully restore the VOR where previous parameterizations could not(Fig. 5). The modification of neural implant algorithms predicted with these equations can be tested with experiments in Fig. 5 for improvements in driving desired firing patterns.

In post-damage and implanted systems, lower levels of activity are expected, as in the implanted vestibular afferents in the data analyzed in this paper⁹; reduced responsiveness to stimulation may occur, as in explanted vestibular afferents compared to in vivo²⁵, or rate of spike-recovery may change as in post-deafness auditory nerve fiber under cochlear implant stimulation³⁴. Whichever case, healthy and damaged neuron parameterizations could be made using the experiments in the text and with a measurement of baseline spontaneous firing rate. Reduced responsiveness to stimulation could be captured in the parameterization of the pulse-spontaneous interaction parameters (p_{sp} , p_{ps}), and differences in channel dynamics due to damage or natural physiological differences in channel types used to drive APs in other systems could be captured with adjustments of pulse-pulse parameters (t_p , p_{pb} , etc.). ...

Our findings suggest several possible improvements to neural implants, even considering the mixed effects of pulses on neuronal populations with natural levels of diversity. A hardware solution that is already under development³⁵ would be to use high-density electrode devices and small amplitude stimuli that are capable of targeting individual neurons. Our study of pulse parameter effects also indicates a number of algorithmic improvements. One inference observable from low pulse amplitude simulations (i.e. Fig. 4a $I < 45\mu A$) is that the PFR would be highly linear for all spontaneous activity levels but with a low slope. Thus, a high-rate low-amplitude stimulation parameterization may induce nearly linear modulation that can induce the upper range of firing rates seen in the system (i.e. 1000 pps producing 500 sps). Additionally, using more complex optimization strategies to find parameters that best co-activate neurons with a range of spontaneous activity levels may be another useful way to use our equations. Still, characterizing a large number of densely packed neurons may be intractable presently, and, especially in highly interconnected areas, such as parts of cortex, time-varying inputs may be difficult to account for. Another potential solution indicated by our study would be to eliminate spontaneous activity or inputs from other areas. For example, one could use gentamicin to ablate vestibular hair cells³⁶, in the case of a vestibular prosthesis, or site-specific channel blockers in cortex. This would make neurons easier to drive with consistency throughout the population because it eliminates pulse-spontaneous interactions and leads to a larger inducible firing range (Figure 4a, Figure 5h). Highly inter-connected regions may remain difficult to characterize and isolate in this way.” (p.15-16)

We agree that adding these and other related works across systems improves the manuscript.

3. With respect to the clinical applicability, the model and the physiological data are describing electrical stimulation from a vestibular prosthesis that is implanted in a healthy animal. There is an extensive literature on how short-term and long-term deafness leads to changes in the response of auditory nerve fibers to stimulation from a cochlear implant. I am not aware of similar studies in the case of vestibular pathology, but it is likely that analogous changes occur in vestibular nerve fiber responses due to damage to vestibular hair cells and subsequent changes in the connected nerve fibers. In addition, as the authors point out, in a population of nerve fibers receiving independent synaptic input, the ability to pattern the electrical stimulation to optimize responses might be greatly limited by the great range of patterns of synaptic input those fibers receive, and for a clinical vestibular prosthesis, there may be no practical way of measuring what synaptic input the population of fibers is receiving. These issues need to be discussed in the manuscript.

We thank the reviewer for bringing up a need to discuss limitations that affect all neural implant efficacy and the applicability of the algorithmic changes discussed in this paper in the context of pathology and other neural systems. These are both important points that we expanded on in the revised Discussion in the section copied in response to comment 2 in response to clinical applicability of the equations.

Specifically, we added the following about damage:

“In post-damage and implanted systems, lower levels of activity are expected, as in the implanted vestibular afferents in the data analyzed in this paper⁹; reduced responsiveness to stimulation may occur, as in explanted vestibular afferents compared to in vivo²⁵, or rate of spike-recovery may change as in post-deafness auditory nerve fibers under cochlear implant stimulation³⁴. Whichever case, healthy and damaged neuron parameterizations could be made using the experiments in the text and with a measurement of baseline spontaneous firing rate. Reduced responsiveness to stimulation could be captured in parameterization of the pulse-spontaneous interaction parameters (p_{sxp} , p_{pls}), and differences in channel-dynamics due to damage or natural physiological differences in channel types used to drive APs in other systems could be captured with adjustments of pulse-pulse parameters (t_p , p_{pb} , etc.).”

We discuss limitations and solutions and specifically interconnectivity in the paragraph that starts:

“Our findings suggest several possible improvements to neural implants, ...Highly inter-connected regions may remain difficult to characterize and isolate in this way.” (p.15-16)

4. Finally, the main equation describing the model, Eq. 5 in line 517 of the manuscript, is incorrect. All of the currents on the righthand side of the equation should be in the numerator, not the denominator, and the voltage-gated ion channel currents and the synaptic current should all have a minus sign in front of them. I believe this is a typesetting error, rather than being an error with the model implementation itself, since the model behavior is as expected. I think that this typesetting error has occurred in transcribing the corresponding equation from the authors' previous modeling article (Steinhardt & Fridman, 2021). In that previous paper, the membrane equation is given correctly on page 2 of the supplemental material, but on page 1 in Eq. 1 of the supplemental material, the minus signs are missing, and the equation is typeset ambiguously using the notation $1/(CmS)(INa + IKL + IKH + I_{leak} + I_{epsc} + I_{stim})$, where the 1/ only corresponds to the first term in parentheses, but in the manuscript under review the second term with the currents has erroneously been placed in the denominator as well. This needs to be corrected.

We thank the reviewer for noting this formatting error in Eq. 5. We have changed it to the form suggested to reflect the way the simulation was run. We also rechecked the other equations in the text for additional scripting errors.

Reviewer #2 (Remarks to the Author):

I read this paper with interest, the authors describe the effects of “pulsatile” stimulation on vestibular axons and try to describe effects that occur at axonal level and that impact the neural output against the naively assumption of $F=R$. Using computational biophysics, they study the complex effects that affect firing rates for different rates and pulse amplitudes with and without the presence of spontaneous neural activity and then they show that recordings of single vestibular fibers do show some of the properties they found. While I think this work is very interesting and timely, I think it fails to provide clear message and example of what this means for stimulation systems and how we should change the way we designed them other than “increase the number of contacts”. Therefore, I think the paper requires some re-thinking to be able to nail down the real impact of the results.

Below my comments.

We thank the reviewer for their positive feedback regarding the significance and timeliness of our study. We have made substantial revisions to the manuscript to address each of their comments as detailed in our point-by-point responses below.

Novelty and literature

1. While I understand that the authors are focused on vestibular stimulation and therefore are knowledgeable of the literature in that field. Since they pitch the paper as to studying fundamental principles that apply “in general” in neurostimulation they have to know that these pulse-pulse and pulse-spontaneous activity interactions has been studied already in other applications. Specifically I would suggest them to look at the papers of Formento et al Nature Neuroscience 2018 and the preprint from Steve Prescott lab (<https://doi.org/10.1101/2023.01.10.523167>). None of this removes importance from their findings, but I think they must be put in context. For example, the explanation of the pulse-pulse interaction is essentially describing pulse skipping, or the pulse-spontaneous activity, is describing cancellation of ongoing natural activity, with relationships that have been previously observed and explored in computational models.

We agree with the reviewer that is important to emphasize the generality of the principles of pulsatile stimulation discussed in this paper and, to do so, that it would be helpful to point to previous experiments and simulations that show evidence of these pulse effects across systems.

We thank the reviewer for bringing to our attention these recent papers. In our revised Discussion, we now provide a broader review of work in other systems, including references to these studies: “The resulting PFRs resemble pulse effects demonstrated across neural systems: high-frequency facilitation (row 1) has been observed in auditory nerve fibers^{27,30}; the PPB effect that leads to a bend in the PFR (row 2) has been observed in auditory nerve fibers²⁵ and dorsal column axons^{22,27,30}; amplitude-dependent growth of firing rates has been observed in the auditory nerve³²; high amplitude block is observed in the sciatic nerve (row 3)³¹; pulse-spontaneous additive and blocking effects has been observed in experiments on hippocampal neurons³³, auditory fibers²⁸ and spinal cord proprioceptive fibers²¹ (Fig. 4a). These similarities further support our hypothesis that there is a large source of shared variability in effects of pulses in clinical applications due to pulses driving axonal channel dynamics to unnatural states.” (p. 15)

In addition, to address the reviewer’s concern, we have revised the Introduction to place our work more clearly in the context of the history of detailed biophysical modeling across neural implant use-cases, highlighting working in DBS, SCS, and cochlear implants:

“Neural engineers have explored the factors that impair neural implant performance using detailed biophysical models that include neuron-specific channels, ion densities and physiology¹²; such modeling has been especially pertinent because stimulation artifacts and technological limitations often prevent direct observation of neural responses during therapeutic intervention. Particularly, the deep brain stimulation (DBS) field has used this approach to understand the impact of parameters such as pulse waveform, electrode orientation, and tissue properties on neural activation¹³. Successes in this field have led to the use of patient-specific modeling as a popular clinical approach for finding patient-specific

stimulation parameters that improve the performance of a variety of implants^{11,14,15}. These parameterizations however do not account for another essential feature of neural responses: the neuronal firing pattern over time.

Producing consistent, interpretable neuronal firing patterns in real-time is a critical factor in restoring function, particularly in sensory systems, where the natural firing patterns carry information about time-varying sensory input signals to the brain. Neural implants therefore employ algorithmic mappings that determine the stimulation parameters needed to evoke the desired neuronal firing pattern. Standard stimulation strategies include fixed-amplitude pulse rate modulation^{16,17} and fixed-rate pulse amplitude modulation¹⁸, where in both cases the fixed parameter is set at a high level. An assumption that is generally inherent to these strategies is a one-to-one mapping between each stimulation pulse and neuron firing¹⁹. However, experimental observations and mathematical modeling^{10,20–22} have identified effects that can lead to time-varying differences in firing rate, including facilitation and blocking^{10,20}, especially when combined with ongoing spontaneous (natural) firing activity. We propose that these effects, which lead to complex relationships between pulse parameters and neural activation, are a common reason for the limited restorative efficacy of neural implants.” (p. 1-2)

Figures

2. The figures are difficult to understand, and legends do not help. Indeed, they aren't legends, but further description of the results. The legends should describe the elements represented so that one can understand what's being represented.

We thank the reviewer for pointing out the need to improve the utility of our figure legends. We agree that further description and clearer panels were necessary. We have thoroughly revised the manuscript to point to specific features of each of the panels in each figure and added further descriptive text to each figure, directly mentioning line/marker colors. We have also broken some panels (such as Fig. 3b) up for clarity. We have also revised supplemental figures with this in mind. To elucidate the pulse effects in subpanels more clearly, as, especially in displays of pulse-spontaneous effects, there are often multiple layers of effects co-occurring, we added Supplemental Figure 1 which isolates pulsatile effects referred to in Figs. 2-3 for easier visualization and shows how parameters control those features.

Examples include a revision of Fig. 2 caption:

“Fig. 2: Pulse-pulse interactions. Interactions go through three stages: a) facilitation, b) addition, and c) suppression. In each stage, as current increases (indicated by color bar in panels a-c) the pulse-pulse interactions change, changing in the PFR. The parameters governing the dominant effect are shown on the line graphs below each figure. Colors of markers on line graphs are matched to I in color bars. The lines $F=R/n$, $n=1,2,3\dots$ are shown with dashed lines on graphs to compare to local slope of the PFR. d) Examples of each effect are shown with the pulse train on the top. The duration of the full-block window t_b is marked in blue on the pulse train. The voltage trace is shown with color matched to the current level that produces them in the PFR plots in a-c. The dynamics of the h-gate (green) and m-gate (grey) of the sodium channel are shown. The dynamics can be used as a correlate of the axon state. e) A close-up of the PFR for $68 \mu A$ (the I where pulse-pulse block was demonstrated in d2 and d3.1, lime green from b). R_b (filled circle) and R_{pb}^1 (open circle) are marked on the PFR. The R_s at which effects from d are shown (thin dash line), as well as example $R=100$ pps where partial block occurs (thick dash line). f) Partial block is illustrated overtime at $68 \mu A$ in the same format as d. In m-gate (grey), reduced channel activation can be seen (marked with circles) with each pulse, leading to 1/5 pulses per sequence being blocked ($F=0.8R$ in e). g) Comparison of the rules for each pulse-pulse blocking effect for the same parameters t_b , $p_{bp}^{1/2}$, and $k_{bp}^{1/2}$ are shown and colored by the implemented rule.” (p. 5)

And text edits such as in the Results:

“Although EPSC timing and thus the subset of EPSC events that generate APs are stochastic, because of their frequency compared to pulses, interactions can be estimated to occur with approximately uniformly distributed EPSCs (Fig. 3b yellow line, histogram). For a given I , there is some $t_{pxs}(I,S)$ after a pulse for which a pulse blocks EPSCs from becoming APs (purple), an analogue of t_b . As R increases, the ratio of $t_{pxs}(I,S)$ to the inter-pulse interval ($1/R$) increases to 1; we capture this effect with $p_{pxs}(I,S)$, the probability that a pulse blocks spontaneous APs, where once $p_{pxs}(I,S)=1$, each pulse blocks all spontaneous APs in between (Fig. 3c top).

At the same time, the ever-present EPSCs create a constant resistance of the axon to pulses, captured by p_{pls} , the probability that a pulse produces an AP given the spontaneous activity level. When I is low,

$p_{p_{xs}}=0$ and $p_{p_{ls}}=0$. As I increases, pulses are sufficient to overcome the EPSC activity and eventually block all spontaneous APs, so $p_{p_{xs}}$ and $p_{p_{ls}}$ go to 1 (Fig. 3c-d, Supp. Fig 2 for changes with I and S)."

Validity and generalization of the mathematical model

3. The authors present their model capturing general principles to describe the effects on Firing rates of the different phenomena that can occur at the axon level. While I agree that these can be described with equations, that "model" does remain a fit to data. This means that, obviously, the equations are only valid for the specific set of simulated axons. As the authors point out indeed, different ion channel distribution affects all these phenomena, so their "model" would have to be refitted to the specific models of other neurons in order to generalize to different applications. For example, it would not hold for spinal motoneurons that have very slow calcium dynamics.

We thank the reviewer for pointing out that how the equations fitted to vestibular afferents in the main body could be fitted to neurons from other systems was not stated clearly in our previous version of the Discussion. We significantly revised our Discussion to address this point and other limitations and applications of this work as discussed in response to comments 4-6 below.

The new Discussion now starts by referencing evidence in vestibular afferents and other systems, such as the proprioceptive fibers in Formento *et al.* of the pulsatile effects we characterized to support our expectation that these models can apply to other systems:

"The resulting PFRs resemble pulse effects demonstrated across neural systems: high-frequency facilitation (row 1) has been observed in auditory nerve fibers^{27,30}; the PPB effect that leads to a bend in the PFR (row 2) has been observed in auditory nerve fibers²⁵ and dorsal column axons^{22,27,30}; amplitude-dependent growth of firing rates has been observed in the auditory nerve³²; high amplitude block is observed in the sciatic nerve (row 3)³¹; pulse-spontaneous additive and blocking effects has been observed in experiments on hippocampal neurons³³, auditory fibers²⁸ and spinal cord proprioceptive fibers²¹ (Fig. 4a). These similarities further support our hypothesis that there is a large source of shared variability in effects of pulses in clinical applications due to pulses driving axonal channel dynamics to unnatural states." (p. 15)

We then add a discussion of how to apply them to other systems with different channel properties: *"In post-damage and implanted systems, lower levels of activity are expected, as in the implanted vestibular afferents in the data analyzed in this paper⁹; reduced responsiveness to stimulation may occur, as in explanted vestibular afferents compared to in vivo²⁵, or rate of spike-recovery may change as in post-deafness auditory nerve fiber under cochlear implant stimulation³⁴. Whichever case, healthy and damaged neuron parameterizations could be made using the experiments in the text and with a measurement of baseline spontaneous firing rate. Reduced responsiveness to stimulation could be captured in the parameterization of the pulse-spontaneous interaction parameters (p_{exp} , p_{pls}), and differences in temporal channel dynamics due to damage or natural physiological differences in channels used to drive APs in other systems could be captured with adjustments of pulse-pulse parameters (t_p , p_{pb} , etc.)."* (p. 15)

Overall, we expect these rules would "hold for spinal motoneurons [and other neurons] with [different] dynamics" and other properties. For example, in the case of a spinal motoneuron with slow dynamics, changing the value t_b that determines the pulse rate at which pulse-pulse blocking bends would be predicted to create a tuned version of the rules that captures pulsatile effects in those neurons.

Relevance for neurotechnology applications

4. While I really think that this relationship must be explored the paper falls short at providing a clear example of how their model is going to help. Specifically, the authors report in Figure 5 that their equations can be used to produce more accurate sinusoidal firing rates. However, nowhere in the paper is shown what exact stimulation protocol is the model suggesting. This is very

important because the output could be unfeasible. The authors described it as “still feasible” but provide no data to support that statement.

We thank the reviewer for pointing out this key point. We have revised the manuscript and now provide a specific example of how our model is going to help the field in our discussion of Figure 5 and former Supp. Fig. 4 now 7, which directly shows a simulation in which our implementation leads to improved control of firing rate over time in vestibular afferents compared to standard vestibular implant algorithms and in the case of PRM and PAM applications:

“Next, we assess how the pulsatile stimulation effects shown in this paper could alter the fidelity of desired firing patterns during standard stimulation paradigms. We take the case of vestibular prostheses where, standardly, the natural head velocity to firing rate mapping (black dash) is used to generate a target firing rate from detected motion; then, a one-to-one mapping between pulse rate and desired firing rate is used in a pulse rate modulation (PRM) strategy, under the assumption that at a high pulse amplitude, here 250 μ A, each pulse will produce an AP¹⁹ (Fig. 5a-c). With present stimulation algorithms, impaired vestibular ocular reflexes (VORs) are partially restored in the direction of increasing firing rate and less so in the direction of decreasing firing rate from baseline²⁹. These results occur in afferents that have some residual spontaneous activity. We simulate this case in an afferent with spontaneous activity ($S = 31$ sps), receiving PRM to encode a sinusoidal eye velocity (Fig. 5a-c). The predicted head velocity to induced firing rate mapping can be plotted by remapping based on the PFRs at these parameters. Using the one-to-one mapping (purple), the firing rate should not reach the maximum or minimum desired firing rate, and it shows a relative bias towards being able to excite compared to inhibit (Fig. 5b). These responses reflect limitations in VOR observed in animals and humans with vestibular implants²⁹.

The equations described above can also be inverted to predict the optimal pulse rate -in this case the minimum pulse rate- for inducing a desired firing rate. When we do this (see Methods), we see a monotonic encoding of head velocity can be restored using the same range of pulse amplitude and rate parameters (Fig. 5b, blue); it only requires a more complex but achievable modulation strategy (Fig. 5c-d blue). We then simulate the afferent response to each stimulation paradigm and see the predicted limitations in induced firing rate with the one-to-one mapping and the desired firing rate response from the corrected paradigm (Fig. 5c-d, Supp. Fig. 7). Although the equations were derived from 1-second fixed-rate fixed-amplitude pulse trains, we see the rules explain the limitations of the one-to-one mapping and consistently predict stimulation patterns that can produce sinusoids and more complex mixtures of sines with high fidelity from individual afferents of various spontaneous rates with PRM and PAM paradigms (Fig. 5c-d, Supp. Fig. 7c-d). (p. 13)

We agree with the reviewer that the visibility of the protocol used for producing optimal pulse parameterizations over time and a description of how it was parameterized to be “feasible” is a significant aspect of our findings. We have thus revised the Methods section “Pulse Rate and Amplitude Modulation” to explain this in more detail:

“To test how the pulse rules apply to sinusoidal modulation, as used in various prosthetic algorithms, PRM and PAM were simulated with pulse parameters restricted to the range commonly used in vestibular prostheses: pulse amplitudes 0 to 350 μ A and pulse rates between 0 and 360 pps^{5,19,43}. We use a simple optimization strategy, as a demonstration of the applicability of these equations. For PRM, the common vestibular prosthetic strategy, a PFR is generated at the chosen pulse amplitude based on the equations. Then, the lowest pulse rate that produces the target firing rate desired (or the closed firing rate achievable using rms) is selected (Fig. 5a). For PAM, in an analogous manner, the chosen pulse rate is selected, and the pulse amplitude-firing rate mapping is used to select the lowest pulse amplitude that produces the desired firing rate (or the closed firing rate achievable using rms). Potential pulse amplitudes and rates were sampled in steps of 1 μ A and 1 pps. This solution was a simple approach for minimizing energy consumption in either stimulation paradigm. For a moving firing rate prediction in the text, the target firing rate trajectory is sampled at 0.1 ms sampling frequency, and optimal pulse parameters are chosen at each time step. ” (p. 19)

Due to the use of “pulse parameters restricted to the range commonly used in vestibular prostheses” and the time-independence of the equations, we propose that the solutions provided with this code are practical and feasible for use in existing devices.

For more generally how the model is going to help, please see responses to 6 and 7 below.

5. The only additional figure is supplementary figure 7 that shows the effects of different pulse shapes one of which, the “asymmetric pulse shape”, it’s what’s being used in neurostimulators from Medtronic since the 70s.

In response to this comment, we have revised the text around this figure in the discussion to clarify that the last supplemental figure was not introduced as the main solution proposed in this work. This figure was included to address other ways our modeling approach can provide answers beyond changing the existing standard stimulation paradigms with biphasic-charge balanced pulses:

“Additionally, understanding the source of pulse effects, as we do for biphasic pulsatile stimulation here, may help to design novel stimulation waveforms with beneficial effects. For example, we show that the cathodic phase of pulses leads to the blocking effects, and the anodic recovery phase can affect the duration of the evoked spike afterhyperpolarization; using this information, the shape of the recovery-phase of a pulse could be designed to sensitize the axon so that when the next pulse is delivered one-to-one AP induction occurs (Supp. Fig. 10). We can use a similar analysis to that in the text to create equations that capture effects of these pulses.” (p.15-16)

6. So how exactly is the model changing the way we stimulate, if that’s the goal?

The reviewer is correct that this is a major goal of the paper. We appreciate that the reader would benefit from an extended discussion about the implications of this work for improving electrical stimulation more broadly. Accordingly, we have edited our discussion of Fig. 5 to highlight how our algorithm could improve vestibular prostheses (see or response to 4 above). We also extensively revised the Discussion in paragraphs 2-5 to more broadly discuss how our equations can be applied to other systems and other implications of our findings for improving stimulation algorithms and neural implant design.

In the modified Discussion, we first discuss how these equations could be applied to other systems, as discussed above in response to comment 3.

We then discuss possible solutions for applying these equations to systems with more complexity in the section:

“These results imply that our present uses of pulsatile stimulation are not producing coherent local excitation in most cases, due to diversity of baseline neural activation levels. They produce a consistent but unnatural combination of local excitation and inhibition, where the response of a neuron is based on its ongoing level of activity and distance from the electrode site.

Our findings suggest several possible improvements to neural implants, even considering the mixed effects of pulses on neuronal populations with natural levels of diversity. A hardware solution that is already under development³⁵ would be to use high-density electrode devices and small amplitude stimuli that are capable of targeting individual neurons. Our study of pulse parameter effects also indicates a number of algorithmic improvements. One inference observable from low pulse amplitude simulations (i.e. Fig. 4a $I < 45\mu\text{A}$) is that the PFR would be highly linear for all spontaneous activity levels but with a low slope. Thus, a high-rate low-amplitude stimulation parameterization may induce nearly linear modulation that can induce the upper range of firing rates seen in the system (i.e. 1000 pps producing 500 sps). Additionally, using more complex optimization strategies to find parameters that best co-activate neurons with a range of spontaneous activity levels may be another useful way to use our equations. Still, characterizing a large number of densely packed neurons may be intractable presently, and, especially in highly interconnected areas, such as parts of cortex, time-varying inputs may be difficult to account for. Another potential solution

indicated by our study would be to eliminate spontaneous activity or inputs from other areas. For example, one could use gentamicin to ablate vestibular hair cells³⁶, in the case of a vestibular prosthesis, or site-specific channel blockers in cortex. This would make neurons easier to drive with consistency throughout the population because it eliminates pulse-spontaneous interactions and leads to a larger inducible firing range (Figure 4a, Figure 5h). Highly inter-connected regions may remain difficult to characterize and isolate in this way.” (p.16)

We also offer another stimulation protocol to explore:

“These outcomes suggest a potentially exciting direction for improving stimulation algorithms is to focus on neural signatures of coherent population-level encoding as opposed to producing a high fidelity single-neuron response in targeted neurons in the population.” (p. 16)

7. An additional problem with the relevance is whether there is a problem at all. I am a fan of general theoretical papers, but the authors failed to identify a specific flaw in current applications. For example, cochlear implants work well, as they describe, even if the firing rates are all affected by the pulse-pulse problems. I think it is important to pinpoint a specific application in which the fact that F is not R does affect clinical outcomes, then, in that case, it will become very clear why this model helps.

We thank the reviewer for pointing out the need to highlight how the model helps address specific flaws in current applications. We have revised the Results and Discussion around Fig. 5 to more directly explain how our findings could be used to correct stimulation protocols across neural systems. We discuss the specific application of this rule to repairing the VOR in the section mentioned in response to comment 4 above which is shown directly in contrast to a standard paradigm that assumes $F=R$.

We also add in the Discussion:

“We show equations fitted to one-second blocks of fixed rate-fixed amplitude stimulation can predict responses to pulse rate and pulse amplitude modulation sequences and correct them for pulse effects with modulation on the 5-50 ms timescale (Fig. 5, Supp. Fig 7c-d). Corrections produce firing patterns in silico that under healthy neurological conditions could fully restore the VOR where previous parameterizations could not (Fig. 5).” (p.15)

We additionally now provide specific modifications of protocols and ways to apply these equations to other systems in the modified Discussion, as discussed in response to comments 3 and 6 above.

In our original submission, we discussed the fact that “cochlear implants work well” in the context of why population-level encoding may play a role in restoration of function. Yet, we emphasize that the main message of the paper is that all neural implants show a level of deficit in restoration of function and one common cause may be that they do not presently account for the complex pulsatile effects discussed in this paper, leading to a mismatch between intended firing pattern and induced firing pattern in local populations of neurons. To clarify our narrative regarding limitations across systems, in the Introduction we have now added the following text:

“While pulsatile stimulation-based treatments have successfully aided in a range of restorative and suppressive treatments^{1,2}, patient recovery typically remains significantly below normal levels of function.” (p. 1)

We have also revised the section of the Discussion about cochlear implants to “cochlear implants effectively restore speech perception^{36,37}, although cochlear implant users have remaining deficits like other type of implantees^{5,7}, such as lack of tone discrimination or the ability to hear speech-in-noise³⁹” to address potential confusions. (p. 16)

8. All these points could have been addressed in the Discussion section, instead, the discussion is short and broad.

We agree with the reviewer and have made the point-specific edits described above, which we believe have improved the impact of our Discussion.

9. The topics addressed in this work have significant implications and I would have enjoyed a detailed discussion. For example, in the discussion one of the consequence of their work, according to the authors is to increase the number of stimulation channels. But this “solution” is the same that is being proposed since the start of neuroengineering. So, what is the single important impact that knowing how the pulse rate is affected changes future and present solutions? Or could change them? This is not clear right now and the conclusion seems to be only the old obvious ones.

We thank the reviewer for acknowledging the significance of the implications of this work for neuroengineering. As noted above, we agree that the Discussion required a more detailed and specific discussion of the consequence of this work and potential changes to future and present solutions being implemented in devices. The main solution we suggest in this study is to algorithmically change the pulse parameter choice to account for the pulsatile effects discussed in the paper. We discuss this point in detail in the revised manuscript, as discussed above in response to comment 4, and how to apply this solution to other systems directly, as discussed above in response to comment 3.

We also significantly modified the Discussion to provide solutions besides hardware solutions and added explicit suggestions of specific parameter choices and interventions that could be included in treatments to make this option more viable across systems. Details of these changes are discussed above in response to comment 6.

We believe the changes discussed above, based on the recommendation of the reviewer, strongly improve the quality of the manuscript.

10. In conclusion I think the authors are on the right path and their work could be of big impact but it requires a significant revision of the current form to be able to impact neurotechnology practice.

We thank the reviewer for recognizing the potential impact of this work on neurotechnology. We have substantially expanded on the points above with the goal of elucidating this potential impact for the reader.

Minor comments:

11. Fig. 2: It would be informative to include the sodium ion dynamics during spontaneous activity and natural channel perturbations.

We agree that it is informative to show the pulsatile effects compared to spontaneous activity, although it does not easily fit into Fig. 2. We thought it would fit in well in the discussion of how spontaneous activity affects the pulse-pulse rules discussed in Fig.2. So, we add Supplemental Fig. 3, which shows natural spontaneous dynamics and dynamics in the same afferents during different levels of pulsatile stimulation in the style of traces shown in Fig. 2d.

We have also revised the Results section accordingly: “Finally, S also affects the partial block window of pulse-pulse effects. EPSCs act as a level of noise correlated to S , which extends recovery of the axon after pulses, increasing p_{pb}^n (Fig. 3e middle). Spontaneous activity also prevents PDL by causing too much noise for channels to remain in a dynamic loop so that ψ_1 never exceeds 1 (Fig.3e.1). Example traces of the pulse-pulse effects occurring in afferents with different S are shown, as in Fig. 2d, in Supp. Fig. 3” (p. 10).

12. Fig. 2e: For clarity purposes, it would be helpful to show an entire interpulse interval that indicates t_b and t_{pb} .

We thank the reviewer for bringing up the need to clarify how the t_{pb} block works compared to the t_b block. We agree it is helpful and add a panel (Fig. 2f), showing an example trace, and clarify this in the following added section of the Results as well as in discussing the math in the Methods (p. 21):

“The PFR does not transition directly from $F=R/n$ to $F=R/(n+1)$ at $R_b^n = n/t_b$. Instead, the PFR has a bend, where the slope of the PFR decreases smoothly from 1 sps/pps, starting at R_{pb} (open circle), a R less than R_b , to 0.5 sps/pps at R_b (closed circle, Fig. 2e). In this range of R , pulse-pulse partial block occurs. This effect resembles facilitation. Inter-pulse intervals are short enough for refractory effects to build, but these interactions build to one pulse in a sequence of three or more pulses being blocked instead of one in the sequence producing an AP (Fig. 2f, lime green). Pulse-pulse effects arise from voltage changes affecting the opening and closing of a combination of axonal voltage-gated channels, but a correlate of the effect on the axon state can be observed in the sodium channel dynamics. Here, the m -gate reducing with each pulse (grey), shows this building-blocking effect (Fig. 2f circles). Although a sequence of pulses producing partial block may produce a complex pattern of blocked and added APs, we can estimate the effect on average as the probability of the next pulse in the sequence arriving and being blocked gradually decreasing from 0 to 1 between R_{pb} and R_b (Fig. 2e).” (p.6)

Questions:

We thank the reviewer for the following questions that helped make the manuscript clearer.

13. Line 313: shapes like Fig. 3b left? I do not understand what shapes I should be looking at.

In the revised manuscript, we have now split Fig. 3 into two panels and rewritten the discussion of this portion of the figure in the Results and the caption for clarity. Additionally, we added Supp. Fig. 1 to highlight the relationship between parameters and the shapes/features of the PFR, particularly for effects in Fig. 3. In this section we clarify to describe the effect:

“Spontaneous-pulse (SP) blocking is only observed to block up to one pulse per spontaneous AP in this pulse parameter range, leading to the $\max\{-Sp_{p|s}\}$ term in Eq. 3, where at $p_{p|s}=1$, S pulses are blocked. The SP blocking term is scaled by $p_{p|s}$ because the presence of blockable pulses is scaled down but evenly distributed throughout time, leading to a scaled reduction in pulses for all R .” (p. 9)

14. Line 330: reversal of effects? I do not see the reversal of facilitatory effects in the figure.

We have revised the text for clarity and pointed to it in the new Supp. Fig. 1 where this effect is isolated. *“Facilitation is a slight exception in that $p_{psfacil}$ increases with S until a threshold level of spontaneous activity ($S > 60$ sps) above which primarily SP blocking occurs (Fig. 3d left green vs. blue traces and circles, Supp. Fig. 1).” (p. 10)*

15. Line 335: cathodic block? First, time mentioned on the text as distinguishing the anodic and cathodic. I believe it requires an introduction

We agree that the introduction of this term into the Results occurs suddenly. We understand that the SFP rule discussed prior in the text is a form of cathodic block. To avoid using several terms throughout the manuscript and avoid a discussion of the contribution of the anodic and cathodic phase of the biphasic pulse to the blocking effect, we decided it was best to remove that term in this section and refer to this as, *“At high I , the combination of high I pulses and EPSCs together add to create SFP that blocks pulses.” (p. 10)*

16. Line 350: 12 current amplitudes?

Here, we show 12 exemplar PFRs from spontaneous afferents at different pulse amplitudes to demonstrate the change in pulse effects with pulse amplitude, but we fitted fourteen amplitudes. This is now clarified in the revised text: *“...the PFR of the simulation at fourteen current amplitudes across the seven spontaneous firing rate cases. The parameters are then interpolated for the thirty held-out current amplitude conditions across afferents.” (p. 10)*

17. Fig. 4b: why variability in low S ? I would have expected well-defined behaviours

We thank the reviewer for pointing out this potential source of confusion about the fit at low S. We add this point to the Results with the following: *“We note relative variability in fits at low S compared to high. One source of variability is the accumulation of error at the sharp drops during PP blocking (Fig. 2c), due to the parameters of our equations being bounded to keep parameter exploration reasonable. While, at high S, pulses contribute few APs so non-monotonic blocking effects (PPB, SFP, etc.) are low amplitude, and linear PS and SP blocking effects dominate, which are easily fit with linear rules.”* (p.10-11)

18. Line 381: 0.5-1 slopes PDF? Why not 0-1?

We agree that values between 0-1 and even some negative values during the down slopes of a bend are expected. We have added more detail about the predicted and observed slopes throughout the PFR under high and low pulse amplitude conditions in this section of the Results:

“Due to PPB, we expect a higher frequency of slopes of 1, 1/2, 1/3, particularly at low R, and higher frequencies of slopes close to or less than zero due to pulse-spontaneous block, spontaneous-pulse block, and SFP.” (p. 11)

19. Line 394: simulations of head velocity? This information seems to not be provided.

We have edited the text for clarity regarding what is being plotted and why, specifically: *“We take the case of vestibular prostheses where, standardly, the natural head velocity to firing rate mapping (black dash) is used to generate a target firing rate from detected motion; then, a one-to-one mapping between pulse rate and desired firing rate is used in a pulse rate modulation (PRM) strategy, under the assumption that at a high pulse amplitude, here 250 μ A, each pulse will produce an AP¹⁹ (Fig. 5a-c). With present stimulation algorithms, impaired vestibular ocular reflexes (VORs) are partially restored in the direction of increasing firing rate and less so in the direction of decreasing firing rate from baseline²⁹. These results occur in afferents that have some residual spontaneous activity. We simulate this case in an afferent with spontaneous activity (S = 31 sps), receiving PRM to encode a sinusoidal eye velocity (Fig. 5a-c).”* (p. 13)

20. Fig. 5g what is showing that has not been shown earlier?

The reviewer is correct that the plot is not showing entirely new information. Our goal is to emphasize through the plot that neurons of different spontaneous rates would experience very different facilitation, additive, or blocking effects in response to pulses of the same amplitude. We thought this display would make this fact more visualizable. We note it is a replotted in the caption:

“The response of simulated afferents of different spontaneous rates (S) replotted with panel per S to emphasize the difference in the PFR at the same pulse amplitude, as amplitude increases (blue to yellow).”

We acknowledge this panel is less important than other information in Fig. 5 and shrink this panel in the updated version of Fig. 5 which further emphasizes the potential to use our prediction equations to improve stimulation parameterization.

Reviewer #3 (Remarks to the Author):

In this manuscript, the authors describe the use of computational models to explain the variability of outcomes normally observed with electrical stimulation of peripheral nerves, and specifically to vestibular afferents.

1. I recognise the importance of the development of simplified yet rich descriptions of neural response such as the one presented in this work. In particular, mechanisms of response to high-frequency stimulation are of particular interest, and in this manuscript are well described both mathematically and physiologically, in terms of ion channel dynamics. However, there are some important issues of that should be addressed.

We thank the reviewer for their positive feedback regarding the importance of the description of neural responses presented in our paper.

2. Since this is a modelling study, the authors must publicly release the source code implementing the designed equations, with examples of its use, source data and well-detailed instructions for use. It is mandatory both for the results validity, and since as it could be a valuable tool for other researchers. This can be done in public repository of authors' own choice.

We agree with the reviewer about the value of releasing this code and provided a GitHub link at the time of submission, in which we released the code for the equations, the optimization, and figure generation. We also release the simulation data. Code for the analyses and simulations discussed above can be found at <https://github.com/CSteinhardt153/pulsatileDir> and is also available on request to the authors.

3. The model seems to fit the reference data produced by experimental and computational modeling very well. However, given the great freedom in the choice of parameters (mainly, in the manual definition of the various ψ_n and all the “p”s in the spontaneous-pulse interactions), this is not surprising. Therefore, sensitivity analysis of model outcomes w.r.t. these parameters should be performed: this means shuffling parameters in % range of their nominal values, and studying the outcomes stability.

We thank the reviewer for acknowledging the accuracy of the fit of our model to the data. We also agree that it is helpful to show the sensitivity of this fit to the parameters and to clarify how the parameters were chosen and fit for the reader to not give “great freedom” that leads to unrealistic solutions.

In response to this comment, we added a sensitivity analysis which is shown in Supplemental Fig. 3 and discussed in the Results after showing the fit to the simulation:

“We assess the sensitivity of fit to each parameter, revealing that, although each parameter influences the PFR (Supp. Fig. 1), particularly t_b , $p_{pb}^{1/2}$, and p_{pls} have a strong influence on error in the PFR (Supp. Fig. 4); the pulse-pulse parameters affect rms more with no spontaneous activity. However, as S increases, the various pulse-spontaneous parameters have similar levels of influence to other parameters (Supp. Fig. 4).” (p. 11)

We have also revised the text to clarify the optimization procedure, the difference between hand-fitted and unfitted amplitudes, and how the sensitivity was performed as a result in the modified Methods section “Parameterizing fits”:

“The optimal parameterization of the equations is found using patternsearch in Matlab in the “classic” generalized pattern search algorithm mode which requires parameter initializations and the bounds to be set for each parameter. For a subset of fourteen of the PFRs at simulated pulse amplitudes, the starting parameterizations were found by hand for each of the spontaneous rate cases. At S=0 and S=56, three additional Is were sampled, focusing on the transition points to capture the rule transitions accurately ($I \in [30-100]$ and $I \in [150-250]$). The maximum and minimum I cases were included in this group. These fits are referred to as hand-fitted. For the remaining Is, linear interpolation between the fitted Is followed by optimization is used to obtain optimal parameters. This technique was done to increase the chance of optimization finding solutions involving smooth changes in parameter values that reflect the observed

mechanism of AP generation. For fitting details of the parameters, see Supplemental Table 1, and for observation of the parameterization across I and S conditions, see Supp. Fig. 2. Standard rms error is used for optimizing the best fit at each amplitude. Data are fit to the mean of across simulations. The fit is reported for error across each of the ten simulated runs per model. Difference between error levels of fitted and interpolated PFRs is assessed with a paired t-test. Data is all reported as mean rms across repetitions \pm sem.

A sensitivity analysis was performed on the optimized parameterization of the fitted I cases. All optimized parameters were held fixed except for one which was jittered 100 times within a Gaussian range of 10% of the optimized value. The effect on rms between predicted and simulated PFR was then assessed and reported in Supp. Fig. 4. “ (p. 20)

4. Regarding validation against experimental data, the arguments are quite limited. The choice of sparsely sampling the simulated data, which yields non-significance the in KS test, is hardly justifiable. Of course instead there is significant difference when compared to 5000 permutations, it is not a fair comparison. In general, I recommend to report the reference experimental data from Mitchell et al. more thoroughly and to try to perform qualitative or quantitative comparisons in a less convoluted way. What is currently reported in Fig. 4d and used for validation seems quite reductive compared to the experimental dataset.

We agree with the reviewer and now more explicitly explain the motivation for our approach used to validate against experimental data. We also more thoroughly describe how the experimental data helps with that justification. Specifically, we have extensively revised this section of the Results to add new analysis and explain these points:

We more fully describe the data:

“We then test whether these equations reflect observable features of experimentally recorded vestibular afferents. We reanalyze recordings from six afferents from the Mitchell et al. (2016) study, which focused on central adaptation but recorded vestibular afferent responses to pulsatile stimulation at multiple amplitudes. This provided 5 afferent recordings at the maximum safe pulse amplitude and 4 afferent recordings at eighteen pulse amplitudes from 25% to 100% of the safe pulse amplitude range for that electrode position (see Methods, all data in Supp. Fig. 5). The PFRs show non-monotonicities that could be explained by PPB effects, SFP at high Rs, and changes in PFR with I that reflect results of the simulations (Fig. 4d, Supp. Fig. 5a-b).” (p. 11)

In response to this comment, we re-analyzed the data, finding additional ways to draw conclusions. In the revised section we add more detailed justifications of the analyses, additional t-testing for statistics, and additional comparisons of the AUC for activation and violin plots broken down at high versus low I across afferents to support how aspects of the experimental data reflect observations of the simulations:

“The experimental PFRs could be fit with the equations described above. However, the sparsity of pulse rate and pulse amplitude sampling causes multiple parameterizations of our equations to result in a low rms fit, making it unclear which rules shown led to the result. Instead, we use two metrics to assess the presence of the pulse effects in the data that allow data to be pooled across afferents, increasing the sample size for statistical comparisons. The slope between sampled combinations of pulse rate and firing rate (grey dash and circle) (Fig. 4d left) is used as the main metric for assessing the presence of blocking effects. The normalized area under the curve (AUC) for the PFR (Fig. 4d grey filled) is used as a metric of the level of activation (see Methods). Due to PPB, we expect a higher frequency of slopes of 1, 1/2, 1/3, particularly at low R, and higher frequencies of slopes close to or less than zero due to pulse-spontaneous block, spontaneous-pulse block, and SFP. We first compare the presence of all slopes in the data to slopes in the model. To make a fair comparison to the model, we sparsely sample the simulated PFRs and slopes (see Methods), producing PFRs and pulse rate-slope plots that closely resemble those sampled from experimental afferents of matched spontaneous rates (Fig. 4b right, Supp. Figs. 5-6). The probability density functions of the simulated and experimental data show similar clustering around slopes of 0 with peaks forming near 0.5 and 1 sps/pps that occurred at similar pulse rates (Fig. 4e, Supp. Figs. 5-6). The simulated and experimental distributions are not statistically significant (Welch t-test: $t(622)=0.31, p=0.75$; Kolmogorov-Smirnov test: $p_{KS}=0.16$). A Wasserstein distance $W(P_{exp}, P_{sim})=0.239$ indicates curves are close to each other. The Kolmogorov-Smirnov and Wasserstein distance statistics are significantly different than those between the experimental data and slopes derived from 5000

permutations of the pulse rate-firing rate pairings across recordings, further supporting the similarities in the structure of the experimental and simulated PFRs($p=0, p=0.007$, Supp. Fig. 4d).” (p. 11)

We also investigate pulse rate and pulse amplitude effects in the data. Data cannot be pooled by the I delivered at the electrode because the distance between an afferent and the electrode (which is not known in our experiments) affects the current level received by the afferent. We observed that experimental I values were only increased in a range that led to increasing activation(Supp. Fig. 5b-c), so we assume the maximum I (I_{max}) used would be equivalent in our simulation mapping to $250 \mu A > I > 70 \mu A$. With this assumption, we split PFRs into low R ($R < 150$ pps) and high R sections and compare their slopes at low I ($I \leq 0.5 I_{max}$) and high I to look for pulse amplitude-related effects(see Fig.4f-g for all statistics). At low I , low R slopes are < 0.8 , primarily clustering close to zero in the violin plot, which would be expected from both types of facilitation and SP-blocking and not significantly different than at high R (Fig.4f left). At high I , we expect low R slopes to mostly range from 0-1 (excluding the downswing of the bend that may be captured) and high R slopes to cluster at negative or 0 sps/pps. We see this significant difference in the distribution of slopes($t(78)=3.32, p=0.0014$): positive-valued low R slopes(orange) with clustering around 1, 0.5, 0.25-0.33. and 0 that reflects slopes from PPB and primarily zero and negative valued high R slopes with some samples around 0.5 and 0.3 (purple)(Fig. 4f right). The differences in slope at higher I for the low R region of the PFR are highly significant($t(75)=3.23, p=0.0028$). At low I , the normalized AUC of the PFR is not significantly different at low R or high R, but, at high R, both halves of the PFR show significantly more activation, and the high R portion of the graph reaches a range of significantly higher activation levels (Fig.4g, see for statistics). These results reflect changes in the PFRs for $I < 70 \mu A$ versus $250 \mu A > I > 70 \mu A$ in simulations (Fig. 4a). There were not enough afferents to test for spontaneous rate effects, but distributions are shown in Supp. Fig. 6b.” (p. 11-12)

We feel these changes have significantly improved the manuscript.

5. Additionally, the authors could refer to the works they cite in the beginning of Discussion (lines 460-461) or similar studies from neighbouring fields in neurostimulation (there are successful validations cases in PNS neurostimulation and modelling studies).

We agree that it is important to contextualize this work with successful validation cases of these rules across neural systems and models. In response to this comment, we have added more references, including in the PNS to the list of previous observations of these effects across systems: “The resulting PFRs resemble pulse effects demonstrated across neural systems: high-frequency facilitation (row 1) has been observed in auditory nerve fibers^{27,30}; the PPB effect that leads to a bend in the PFR (row 2) has been observed in auditory nerve fibers²⁵ and dorsal column axons^{22,27,30}; amplitude-dependent growth of firing rates has been observed in the auditory nerve³²; high amplitude block is observed in the sciatic nerve (row 3)³¹; pulse-spontaneous additive and blocking effects has been observed in experiments on hippocampal neurons³³, auditory fibers²⁸ and spinal cord proprioceptive fibers²¹ (Fig. 4a). These similarities further support our hypothesis that there is a large source of shared variability in effects of pulses in clinical applications due to pulses driving axonal channel dynamics to unnatural states.” (p. 15)

6. The authors assert that they invert the equations to predict the PFR and then to find optimal parameters. Is this proper analytical inversion or are we talking about numerical solutions?

We agree that how the equations are inverted should be clearly stated for the reader. We did the latter. We clarify this point in the updated “Pulse Rate and Amplitude Modulation” Methods section: “To test how the pulse rules apply to sinusoidal modulation, as used in various prosthetic algorithms, PRM and PAM were simulated with pulse parameters restricted to the range commonly used in vestibular prostheses: pulse amplitudes 0 to $350 \mu A$ and pulse rates between 0 and 360 pps^{5,19,43}. We use a simple optimization strategy, as a demonstration of the applicability of these equations. For PRM, the common vestibular prosthetic strategy, a PFR is generated at the chosen pulse amplitude based on the equations. Then, the lowest pulse rate that produces the target firing rate desired (or the closed firing rate achievable using rms) is selected (Fig. 5a). For PAM, in an analogous manner, the chosen pulse rate is selected, and the pulse amplitude-firing rate mapping is used to select the lowest pulse amplitude that produces the

desired firing rate (or the closed firing rate achievable using rms). Potential pulse amplitudes and rates were sampled in steps of $1 \mu\text{A}$ and 1 pps. This solution was a simple approach for minimizing energy consumption in either stimulation paradigm. For a moving firing rate prediction in the text, the target firing rate trajectory is sampled at 0.1 ms sampling frequency, and optimal pulse parameters are chosen at each time step.” (p. 19)

7. In Fig. 1c, it is not clear if the pulse current of 230 μA used in the simulation corresponds to the experimental one. If not, reasoning should be provided.

We thank the reviewer for pointing out this potential source of confusion for the reader. The pulse amplitude used in the experiment was $200 \mu\text{A}$, as reported in the experimental data for YLD21. This is a slight mismatch from the best fit at the simulated electrode distance of 2 mm away from the afferent, which was originally chosen based on simulations that specified this distance in another vestibular afferent stimulation study. This mismatch could be corrected for by moving the simulated electrode further from the simulated afferent, thus proportionally scaling the pulse amplitudes at which each effect occurs. We did not do this at the time of starting the simulations, but it does not change the results in a way that affected our analysis of how pulse effects change with pulse parameter changes. At either distance, with proportional changes in pulse amplitude, we observe the afferent progress through the same changes in pulse rate-firing rate effects, and we were able to observe and investigate the range of pulse effects from facilitation to full suppression of activity.

The goal of this analysis in Fig. 1 was to introduce the idea that the simulation can replicate non-linearities in the PFR of experimental afferents and thus was a good choice for investigating pulsatile effects. We felt adding this point to the narrative in this first experiment would complicate the story.

Later in the manuscript, to this point, we add two sections that provide information about why the pulse amplitude mismatch discussed above occurs and how it impacts comparison to simulations and experimental outcomes without bringing up this point at the beginning of our Results section.

We added a discussion of how distance from the electrode affects local current changes and therefore responses of afferents, which makes direct comparisons to the simulation difficult without knowing that distance in the extended Results comparing our simulations to the experimental data in Fig. 5:

“Data cannot be pooled by the I delivered at the electrode, because the distance between an afferent and the electrode (which is not known in our experiments) affects the current level received by the afferent...” (p.12)

In the Discussion, we also now highlight how differences in distance from the electrode affect responses of neurons to the same pulses, adding to the observed variability in responses to stimulation:

“This fact coupled to the fact that local neurons are positioned at different distances from the electrode and thus experience different current levels in response to stimulation implies that our present uses of pulsatile stimulation are not producing coherent local excitation in most cases.” (p.16)

8. In Fig. 2. Are the plots in the bottom row meant to be there? They are not referred to in the caption.

We thank the reviewer for pointing out this graphics error. Indeed, they were not meant to be there, and have been removed in this revision.

9. It is not clear what is represented on the right in Fig.3b, is it the distributions of distances in time between spontaneous APs and the previous pulse? It should be made clearer in the caption.

To address this comment, we edited the caption and made this its own separate subpanel for clarity. 3b is now subpanels 3b) and 3c). The new caption reads: *“b) In the time between pulses spontaneous activity is approximately normally distributed (grey). Thus, it is estimated as uniformly distributed (yellow). As pulse rate increases for the same pulse amplitude stimulating the same afferent, the same length of blocking effect $t_{pxs}(I,S)$ (purple) is present, but with a shorter interval between pulses. c) Thus, the probability of a spontaneous AP being blocked by a prior pulse (p_{pxs}) increases linearly with R until it reaches 1, all pulses blocked (top). Meanwhile, for a given amplitude the probability of a pulse being blocked due to spontaneous activity (p_{pis}) is the same as pulse rate*

increases due to equally distributed spontaneous activity (b) (bottom). Two dots at $I = 36 \mu\text{A}$ indicate R and I combination leading to histograms in b. The level of blocking plotted with changes in I (colored lines).” (p. 8)

10. Also in Fig. 3b, why aren't $p_{psfacil}$ and p_{sxp} represented? It could help comprehension. In fact, the form of p_{sxp} is never explicitly reported in the manuscript.

We thank the reviewer for bringing up the need to elucidate how parameters change with R , I , and S and in turn, affect the PFR. We have revised the discussion of these parameters and added a new Supplemental Fig. 1 to highlight how each parameter changes with R throughout the PFR and how it affects the shape of the PFR. Some of this information was already provided in what is now Supp. Fig. 2 which shows the best parameter fit values through the simulation “for changes with I and S ”:

“This picture of increasing interaction as R increases (Fig. 3b) can be used to visualize why $p_{psfacil}$, the probability of facilitation between pulses and EPSCs, also increases linearly with R at low I s. A similar picture applies for p_{sxp} the probability that EPSCs block pulses from becoming APs. Pulses segment time into inter-pulse intervals, and there is a probability within those intervals of EPSC activity capable of blocking pulses occurring just preceding the pulse, leading to the pulses being blocked. These blocking effects that linearly increase with R co-occur for a majority of I s, making them difficult to isolate in the PFR plots. As such, we show the relevant combination of parameters and their scale below plots in Fig. 3d and 3e and the line graphs below to elucidate how I and S affect those parameters separately. Additionally, in Supp. Fig. 1 right, we highlight the effects of $p_{psfacil}$, p_{pxs} , p_{sxp} , and $p_{p|s}$ on features of the PFR as S increases. Each isolated effect is plotted in red over a PFR trace in insets to the right of the main plots for clarity.” (p. 9)

11. In Fig. 3d are reported results for S between 0 and 132 sps, but in the legend is only listed 13 sps.

The text of the caption was revised to clarify this point. Only Fig. 3c (now d) is shown with low levels of spontaneous activity ($S=13$ sps):

“d) Pulse-spontaneous interactions evolve through a facilitation, spontaneous-pulse block, and pulse-spontaneous block zone as pulse amplitude increases which co-exists with the pulse-pulse blocking rules. Parameters ($p_{psfacil}$, p_{pxs} , p_{sxp} , $p_{p|s}$) governing each are shown changing with pulse amplitude for cases where $S=13$ sps. Pulse amplitude colors are on the same scale as in Fig. 2. e) The same categories of rules and their parameters are shown changing with spontaneous firing rate (colors on right). When a parameter is unlisted below, assume all values are zero across shown cases. e.1) The spontaneous-pulse blocking effect suppressing pulse dynamic loops.” (p. 8)

12. In general, a legend or table of used symbols and their explanation would aid the reader, who instead has to continuously fish through the document. Also, it would be interesting to see reported all parameters in function of S , I , R ... as was partly done in Fig. 3b.

We agree with the review that readers would benefit from clear tables explaining the parameters, how they vary with S , I , and R , and how this affects the PFR. We had previously included some relevant information in Supplemental Table 1, which is referenced in the previous version of the text, particularly when the equations are described in the Methods. The equations were parameterized to fit the PFR, the function that relates R to F , for a given S and I value. This felt logical, because pulse amplitude determines the strength of pulses compared to natural channel dynamics and spontaneous activity, and spontaneous activity level needs to be considered. In Supplemental Fig. 1 (now 2), we show surface plots of the relationship between S and I and the optimized parameters.

To address the reviewer's concerns, we now provide more references to these supplemental materials in the text, such as: “...there is no significant difference in the fit of the parameterized and interpolated conditions, indicating smooth, precise parameterizations could be found (Fig. 4b, Supp. Fig. 2).” (p. 9)

In addition, in the revised manuscript, we have now added the new Supplemental Fig. 1 to further highlight how pulse effects change with I and S and isolate how the parameters in the text affect major features of the PFR. We mention it throughout the explanation of the pulse rules in the results:

“For a summary of how each parameter changes with pulse amplitude and affects the PFR see the left side of Supp. Figure 1.” (p. 7)

“Additionally, in Supp. Fig.1 right, we highlight the effects of $p_{psfacil}$, p_{pxs} , p_{sxp} , and $p_{p|s}$ on features of the PFR as S increases. Each isolated effect is plotted in red over a PFR trace in insets to the right of the main plots for clarity.” (p. 9)

We feel this significantly clarifies the relationship between pulse parameters, S and I , and the shape of the PFR.

The equations should be recalled properly between results and methods: E.g., 3 and 16. It can be confusing especially if they are formulated differently.

We thank the reviewer for noting how changing the order of terms in equations can be unnecessarily confusing for the readers. Equations were reviewed and changed to maintain the same nomenclature and ordering across sections. Both versions now read:

“ $F_{ps} = p_{psfacil}R + \max\{-Sp_{p|s}, -p_{p|s}p_{sxp}R\} + \max\{-S, -p_{pxs}(R - R_{pxs})\}$ ” (p. 9, 23)

13. Lines 331 to 334 are unclear and a parallel with Fig. 3d middle is not evident. The bend does not seem to appear at R_{knee} . Also, because p_{sxp} in function of S is never reported, it is unclear how R_{knee} should behave.

We have revised our discussion of this figure and the relationship between the parameters and features of the graph in the text and legend. As noted in response to 12, between Supp. Table 1 and new Supp. Fig. 1 there is more provided information on the parameterization of p_{sxp} and how this affects the PFR.

We also particularly expand on why R_{knee} may not be observed in PFRs at high S in the added text:

“The point where R_{knee} would have been visible may not be present in PFRs at high S (as in at $S=132$ sps, $I=108 \mu A$). This is due to the combination of the low increase in firing rate with pulses (F_{pp}) and the strong blocking effects blocking all addition of pulses. Mathematically, this is captured in the $\max\{-Sp_{p|s}\}$ term that describes the observed limitation to blocked APs. The knee could still be predicted as it is in Fig. 3e.” (p. 10)

14. Also, in Fig. 3d middle, it is not clear why $\Delta F = S$, should it be placed on Fig. 3d right with a negative sign?

We agree this choice is confusing for the reader. When we added an S to the graph, it was thought of as a reduction of S , a positive drop. We have corrected this to $-S$, meaning reduced F for clarity in this revision.

15. Regarding the testing of accuracy (lines 349-257), it is not clear how and where these 10 amplitudes for parametrization and 10 test amplitudes were selected, among the ~30 steps between 0 and 360 μA ? Related to this and Fig. 4b, it is not clear on what pairs the paired t-test was applied.

We agree and, as discussed above, we have revised our Results section to improve clarity:

“We test the accuracy of these equations by parameterizing them with values that best minimized the rms error between the PFR of the simulation at fourteen current amplitudes across the seven spontaneous firing rate cases. The parameters are then interpolated for the thirty held-out current amplitude conditions across afferents. We find that the equations (red) closely approximate the complexity of the PFRs across conditions (Fig. 4a blues)” (p. 10)

We have also revised the Methods to more clearly explain how they were selected:

“For a subset fourteen of the PFRs at simulated pulse amplitudes, the starting parameterizations were found by hand for each of the spontaneous rate cases. At $S=0$ and $S=56$, three additional I s were sampled, focusing on the transition points to capture the rule transitions accurately ($I \in [30-100]$ and $I \in [150-250]$). The maximum and minimum I cases were included in this group. These fits are referred to as hand-fitted. For the remaining I s, linear interpolation between the fitted I s followed by optimization is used to obtain optimal parameters.” (p. 20)

We thank the reviewer for catching this typo. In Fig. 4b the t-test is applied per spontaneous rate condition between the fitted and unfitted data. An *unpaired* t-test was performed, as there were different numbers of amplitudes in the two groups. The caption now reads: "*Unpaired t-test showing no significant difference between the two (n.s.). Exact values in Supp. Table*" (p. 12). We corrected this error throughout the text and now report degrees of freedom and t-statistical in all cases.

16. Line 360, The count of afferents is not clear, six afferents, 5 and 4...?

We agree and have revised the text, specifically:

"We reanalyze recordings from six afferents from the Mitchell et al. (2016) study, which focused on central adaptation but recorded vestibular afferent responses to pulsatile stimulation at multiple amplitudes. This provided 5 afferent recordings at the maximum safe pulse amplitude and 4 afferent recordings at eighteen pulse amplitudes from 25% to 100% of the safe pulse amplitude range for that electrode position(see Methods, all data in Supp. Fig. 5). The PFRs show non-monotonocities that could be explained by PPB effects, SFP at high Rs, and changes in PFR with I that reflect results of the simulations (Fig. 4d, Supp. Fig. 5a-b)." (p. 11)

17. In Fig. 4d it is unclear what is represented in the right column with respect to the left column.

We have revised the text to clarify this point in the text and caption for Fig. 4:

"d) Experimental recordings from Afferent 1 (that the main simulation in the text was based off). (left) PFR at single I with slope highlighted on the PFR and corresponding slope values shown below on pulse rate-slope plot (grey dot). AUC of firing rate at low R (R<150 pps) also shown (grey fill). (right) PFR and pulse rate-slope plot at increasing Is for the same afferent." (p.12-13)

18. Much more convincing is the perspective application of the identified equations in the optimization of stimulation paradigms to obtain desired firing rates, which has considerable practical implications.

We thank the reviewer for recognizing the importance of the application of the equations created in this manuscript for optimizing stimulation paradigms. In the revised manuscript, we now further emphasize this point throughout the Introduction:

"An advantage of this approach is that resulting equations can be inverted and integrated into real-time devices to correct for complex effects of pulses on firing rate in a computationally efficient way, improving our ability to precisely control neural firing rate over time." (p.2)

and Discussion:

"We show equations fitted to one-second blocks of fixed rate-fixed amplitude stimulation can predict responses to pulse rate and pulse amplitude modulation sequences and correct them for pulse effects with modulation on the 5-50 ms timescale(Fig. 5, Supp. Fig 7c-d). Corrections produce firing patterns in silico that under healthy neurological conditions could fully restore the VOR where previous parameterizations could not(Fig. 5)... healthy and damaged neuron parameterizations could be made using the experiments in the text and with a measurement of baseline spontaneous firing rate" (p.15-16)

Minor comments:

We thank the reviewer for carefully reading and providing these minor comments.

19. Line 31, applications. Changed.

20. Line 92, "spontaneous evoked" seems an oxymoron. I would see it as "evoked" unless argued differently. Line 111, to match

We clarify this by adding the following in the Introduction, "*Throughout the text, we use the term spontaneous to distinguish naturally occurring activity (excitatory-post-synaptic potential (EPSPs) and ESPC-induced spiking) from pulse-induced spiking.*" (p. 3)

21. Line 219, it is unclear what the term "bend" exactly refers to, the points of decrease in FR during PR increase?

Thank you for pointing this out. We expanded the section that introduces the pulse-pulse block rules and

specifically added, "The PFR does not transition directly from $F=R/n$ to $F=R/(n+1)$ at $R_b^n = n/t_b$. Instead, the PFR has a bend, where the slope of the PFR decreases smoothly from 1 sps/pps, starting at R_{pb} (open circle), a R less than R_b , to 0.5 sps/pps at R_b (closed circle, Fig. 2e)." (p.6)

REVIEWERS' COMMENTS

Reviewer #1 (Remarks to the Author):

The authors have generally done a good job of responding to my concerns with the original manuscript.

A few issues with the revised manuscript are:

- p. 2, lines 58-59: I do not agree with the statement: "An assumption generally inherent to these strategies is a one to-one mapping between each stimulation pulse and neuron firing." Designers of stimulation strategies for neural prostheses are typically aware of neural refractoriness and spike-rate adaptation and how they may affect the resulting stimulus encoding.

- p. 3, line 98: The nesting of parentheses in this sentence should be avoided.

- p. 3, lines 114-115: Again, I do not think it is fair to attribute the intuition that F should be equal to R to neural prosthesis designers, when refractoriness and adaptation are generally widely known neural behaviors.

- p. 12, lines 516-518: I would caution against proposing to kill off hair cells in order to remove afferent spontaneous activity, as hair cells could also be providing important neurotrophins to afferent fibers, keeping them alive and regulating their excitability (e.g., see [https://doi.org/10.1016/S0079-6123\(03\)46017-2](https://doi.org/10.1016/S0079-6123(03)46017-2))

- p. 9, lines 415-426: It should be stated more clearly here that the rule-based stimulation strategy is being applied individually to each model afferent with a known spont rate. In a real clinical application, it is going to be difficult to estimate the spont rate of each afferent, and as discussed in lines 503-505, a true implementation of this strategy will only be possible when we have a neural prosthesis that can independently stimuli individual neurons, which is still some ways away.

- p. 14, line 554: The formulation of Eq. (5) is improved, but the ionic currents should all have a negative sign in front of them, since a positive ionic current is an outward current that tends to hyperpolarize the membrane, not depolarize it.

Reviewer #1 (Remarks on code availability):

The code was not accessible via the given link.

Reviewer #2 (Remarks to the Author):

I am impressed by how well the authors incorporated my comments and significantly improved their manuscript.

I truly liked this new sentence:

"These outcomes suggest a potentially exciting direction for improving stimulation algorithms is to focus on neural signatures of coherent population-level encoding as opposed to producing a high fidelity single- neuron response in targeted neurons in the population"

Sometimes there is art in this work, nobody ever gives credit for this, but I will. That is an elegant sentence that captures a very important and modern concept in neuroscience in an effective and succinct way.

congratulations on your work

Reviewer #3 (Remarks to the Author):

The GitHub link is not accessible, it might be a private repository. Please make it available it before possible publication.

Regarding the repository, GitHub is a passable choice, but I would recommend to publish it also on a DOI-minting persistent repository such as Zenodo or similar.

REVIEWERS' COMMENTS

Reviewer #1 (Remarks to the Author):

The authors have generally done a good job of responding to my concerns with the original manuscript.

A few issues with the revised manuscript are:

- p. 2, lines 58-59: I do not agree with the statement: "An assumption generally inherent to these strategies is a one to-one mapping between each stimulation pulse and neuron firing." Designers of stimulation strategies for neural prostheses are typically aware of neural refractoriness and spike-rate adaptation and how they may affect the resulting stimulus encoding.

We agree that this statement was too strong. We revised the section as follows to address this comment:

"Standard stimulation strategies include fixed-amplitude pulse rate modulation^{16,17} and fixed-rate pulse amplitude modulation¹⁸, where the fixed parameter is set at a high level in both cases. An assumption inherent to these fixed-parameter strategies is a consistent linear mapping between the number of stimulation pulses and neuronal firing¹⁹. However, experimental observations and mathematical modeling^{10,20-22} have identified effects that can lead to time-varying differences in firing rate, including facilitation and blocking^{10,20}, especially when combined with ongoing spontaneous (natural) firing activity. We propose that these effects, which lead to complex relationships between pulse parameters and neural activation, are a common reason for the limited restorative efficacy of neural implants" (p.2, line 56-64).

- p. 3, line 98: The nesting of parentheses in this sentence should be avoided.

We have revised this sentence to avoid a nested parentheses as :

"Throughout the text, we use the term spontaneous to distinguish naturally occurring activity, meaning excitatory post-synaptic currents (EPSCs) and ESPC-induced spiking, from pulse-induced spiking." (p.3, line 98)

- p. 3, lines 114-115: Again, I do not think it is fair to attribute the intuition that F should be equal to R to neural prosthesis designers, when refractoriness and adaptation are generally widely known neural behaviors.

We have revised this sentence to clarify that this F=R relationship prediction arises from an assumption that at suprathreshold levels every pulse produces an AP:

"Based on the intuition that a suprathreshold pulse (80% of the level of facial twitch) will induce an AP, at suprathreshold Is, the pulse rate-firing rate relationship (PFR) is expected to be F=R at all Rs."

- p. 12, lines 516-518: I would caution against proposing to kill off hair cells in order to remove afferent spontaneous activity, as hair cells could also be providing important neurotrophins to afferent fibers, keeping them alive and regulating their excitability (e.g., see [https://doi.org/10.1016/S0079-6123\(03\)46017-2](https://doi.org/10.1016/S0079-6123(03)46017-2))

We thank the reviewer for noting this possible confound to killing off hair cells. We have revised the manuscript to instead suggest one of various methodologies for neural silencing:

"Another potential solution indicated by our study would be to eliminate spontaneous activity or inputs from other areas. For example, one could use site-specific channel blockers or other neuronal silencing techniques³⁶. This would make neurons easier to drive with consistency

throughout the population because it eliminates pulse-spontaneous interactions and leads to a larger inducible firing range (Figure 4a, Figure 5h).”

- p. 9, lines 415-426: It should be stated more clearly here that the rule-based stimulation strategy is being applied individually to each model afferent with a known spont rate. In a real clinical application, it is going to be difficult to estimate the spont rate of each afferent, and as discussed in lines 503-505, a true implementation of this strategy will only be possible when we have a neural prosthesis that can independently stimuli individual neurons, which is still some ways away.

We agree with this observation, and we extensively cover this point in the Discussion as well as the paragraph above lines 415-426. As per the Reviewer’s suggestion, we clarified this point in this paragraph.

“...Under the idealized assumption of similar neuronal activity across neurons, the monotonic encoding of head velocity can be restored using the same range of pulse amplitude and rate parameters(Fig.5b, blue); it only requires a more complex but achievable modulation strategy(Fig. 5c-d blue)... ”

- p. 14, line 554: The formulation of Eq. (5) is improved, but the ionic currents should all have a negative sign in front of them, since a positive ionic current is an outward current that tends to hyperpolarize the membrane, not depolarize it. We apologize for the error persisting. We have now verified that the equation is in the correct format:

“The membrane potential (V) varies as:

$$\frac{dV}{dt} = \frac{1}{(C_m S)} (-I_{Na} - I_{KL} - I_{KH} - I_{leak} + I_{epsc} + I_{stim})”$$
 (p.14, line 554)

Reviewer #1 (Remarks on code availability):

The code was not accessible via the given link.

We apologize for this issue. The github link should now be functional, and code is also available on request by email from the first author.

Reviewer #2 (Remarks to the Author):

I am impressed by how well the authors incorporated my comments and significantly improved their manuscript.

I truly liked this new sentence:

“These outcomes suggest a potentially exciting direction for improving stimulation algorithms is to focus on neural signatures of coherent population-level encoding as opposed to producing a high fidelity single- neuron response in targeted neurons in the population”

Sometimes there is art in this work, nobody ever gives credit for this, but I will. That is an elegant sentence that captures a very important and modern concept in neuroscience in an effective and succinct way.

congratulations on your work

Thank you for your kind words. We feel your suggestions greatly improved the quality of the manuscript, and we appreciate your time and effort.

Reviewer #3 (Remarks to the Author):

The GitHub link is not accessible, it might be a private repository. Please make it available it before possible publication.

We have confirmed that the GitHub repository is now public. The code will also always be available on request to the corresponding author.

Regarding the repository, GitHub is a passable choice, but I would recommend to publish it also on a DOI-minting persistent repository such as Zenodo or similar.

Thank you for this suggestion. We have elected to use GitHub to accommodate future updates to the code with additional features in follow-up work. Per your suggestion, we have uploaded our dataset to Zenodo, as well.